# Impact of High-Efficacy Therapies for Multiple Sclerosis on B Cells

**DOI:** 10.3390/cells14080606

**Published:** 2025-04-17

**Authors:** Federica Galota, Simone Marcheselli, Sara De Biasi, Lara Gibellini, Francesca Vitetta, Alessia Fiore, Krzysztof Smolik, Giulia De Napoli, Martina Cardi, Andrea Cossarizza, Diana Ferraro

**Affiliations:** 1Department of Biomedical, Metabolic and Neurosciences, University of Modena and Reggio Emilia, 41121 Modena, Italy; 325706@studenti.unimore.it (F.G.); 224143@studenti.unimore.it (S.M.); 198506@studenti.unimore.it (K.S.); martinacardi20@gmail.com (M.C.); 2Department of Medical and Surgical Sciences for Children and Adults, University of Modena and Reggio Emilia School of Medicine, 41125 Modena, Italy; sdebiasi@unimore.it (S.D.B.); lara.gibellini@unimore.it (L.G.); andrea.cossarizza@unimore.it (A.C.); 3Neurology Unit, Department of Neurosciences, Ospedale Civile di Baggiovara, Azienda Ospedaliero-Universitaria di Modena, 41126 Modena, Italy; vitetta.francesca@aou.mo.it (F.V.); alefiore@unimore.it (A.F.); 4National Institute for Cardiovascular Research, 40126 Bologna, Italy

**Keywords:** multiple sclerosis, high-efficacy therapies, B cells, disease-modifying therapies (DMTs), anti-CD20, Alemtuzumab, Cladribine, Natalizumab, S1P

## Abstract

Multiple sclerosis (MS) is a chronic inflammatory and neurodegenerative autoimmune disorder of the central nervous system characterized by demyelination and neurodegeneration. Traditionally considered a T-cell-mediated disease, the crucial role of B lymphocytes in its pathogenesis, through different mechanisms contributing to inflammation and autoreactivity, is increasingly recognized. The risk of long-term disability in MS patients can be reduced by an early treatment initiation, in particular with high-efficacy therapies. The aim of this review is to provide an overview of the mechanisms of action of high-efficacy therapies for MS, with a focus on their impact on B cells and how this contributes to the drugs’ efficacy and safety profiles. Anti-CD20 monoclonal antibodies, Alemtuzumab, Cladribine and sequestering therapies encompassing Natalizumab and S1P receptors modulators will be discussed and emerging therapies, including Bruton’s Tyrosine Kinase inhibitors, will be presented.

## 1. Introduction

Multiple sclerosis (MS) is defined as a chronic inflammatory and neurodegenerative autoimmune disease that affects the Central Nervous System (CNS), characterized by demyelination with axonal loss, astroglial proliferation and grey matter impairment in genetically susceptible hosts [1].

The pathogenesis of MS is complex and involves both environmental and genetic factors [2]. Traditionally, MS has long been considered a T-cell-mediated autoimmune disease, with autoreactive T cells recognised as the main culprits of neuroinflammation/degeneration and demyelination [3]. After being activated in the periphery, presumably via various infectious agents acting as molecular mimics of CNS-antigens [4], T cells have, indeed, the ability to cross the blood brain barrier (BBB), which was shown to be altered in MS patients [5], and initiate the inflammatory cascade that results in CNS damage [6]. The presence of autoreactive T cells in MS patients has been demonstrated in various studies, particularly Th1 and Th17 cells, which produce pro-inflammatory cytokines such as IFN-γ and IL-17, contributing to myelin damage and neurodegeneration [7]. This evidence also derives from the demonstration that experimental autoimmune encephalomyelitis (EAE), a demyelinating CNS disease similar to MS in both clinical and pathological aspects, serving as a model for studying the human disease, can be triggered by transferring activated myelin-specific CD4+ T cells from mice with EAE into naïve mice [8,9].

Nevertheless, in the last decade, an increasingly well-established body of evidence has shifted our understanding of MS pathogenesis to a critical role for B lymphocytes [10]. Multiple studies have shown that B cells are involved in the pathogenesis of MS by a variety of mechanisms including antigen presentation, cytokine release and antibody production [11,12]. In this regard, it is crucial to mention the phase II study of Hauser et al. [13], showing that targeting CD20+ B-cells specifically could be a successful method in the treatment of relapsing-remitting MS (RRMS) patients, determining a great decrease in the number of gadolinium-enhancing (GdE) lesions in Magnetic Resonance Imaging (MRI) and clinical relapses in comparison to placebo.

The prominent role of B-cells in MS pathogenesis is supported by a number of observations [14], including the presence of IgG oligoclonal bands (OCB) within cerebrospinal fluid (CSF) in the majority of MS patients [15]. Autoantibody production against myelin and other constituent of the CNS is a central task for B cells in MS can mediate direct cytotoxicity via complement activation, resulting in myelin damage [16]. Furthermore, B cells function as antigen presenting-cells (APCs) and effectively present myelin antigens to T cells, maintaining the autoimmune response [17]. B cells can be found in brain lesions from MS patients and in ectopic lymphoid follicles. These follicles, commonly found within meninges [18], might serve to perpetuate immune-driven inflammation and constant autoimmunity inside the CNS. Moreover, B cells present within the CNS secrete TNF-α and IL-6 found in a pro-inflammatory state, exacerbating inflammation and promoting the survival of pathogenic T cells [19], suggesting a complex role where B cells could also regulate the immune response under particular conditions.

This reframing of the disease has important implications for therapeutic strategies, which, up until recently, have been focused more on T cells, leading to a “B-lymphocyte centric view”, where the latter are recognised as main actors and central contributors in the disease development [20].

Indeed, the intricate network of cells and molecules responsible for driving MS pathology indicates that the best treatment should focus on multiple components. The goal of the management of MS is to lower the risk of relapses and the development of disability, according to the guidelines established by the European Academy of Neurology (EAN) and the European Committee for Treatment and Research in Multiple Sclerosis (ECTRIMS [21]. Currently licensed disease modifying therapies (DMTs) have the ability to lower the occurrence and intensity of MS clinical relapses, decrease disease activity (as measured on MRI) and also prevent disability progression by regulating the immune system [22,23]. Considering their efficacy, particularly concerning relapses reduction rate [22], DMTs are generally distinguished into moderate efficacy (ME) DMTs, including Interferon-beta (IFN-β), Dimethyl fumarate, Glatiramer acetate, Teriflunomide, and high efficacy (HE) DMTs, encompassing Ocrelizumab, Ofatumumab, Alemtuzumab, Cladribine, Natalizumab and Sphingosine-1 phosphate modulators [23].

In this review, we will focus on HE DMTs for MS. In particular, we will outline their mode of action, with particular reference to their impact on B cells and how this contributes to their efficacy and safety profiles, encompassing the response to vaccinations.

## 2. Cell-Depleting Therapies

### 2.1. Anti-CD20 Monoclonal Antibodies

#### 2.1.1. Mode of Action and Efficacy

In recent years, increasing attention has been placed on anti-CD20 monoclonal antibodies (mAbs) that mediate B-cell depletion, such as Rituximab, Ocrelizumab, Ofatumumab and Ublituximab. These therapeutic options have demonstrated to be very successful in the treatment of MS patients due to their high efficacy in relapsing forms of MS (RMS) patients, including Clinically Isolated Syndrome (CIS), RRMS and active secondary progressive MS (SPMS), [13,24,25] and in slowing progression in Primary Progressive (PP) MS, although Ocrelizumab is the only one approved and with proven efficacy in the latter clinical form [26,27]. Anti-CD20 antibodies specifically target the CD20 (“Cluster of differentiation 20”) molecule, which is expressed on pre-B cells in the bone marrow, as well as on naïve, memory B cells and early plasmablasts in the lymphoid tissues or germinal centers [28]. Conversely, CD20 is not expressed by most plasmablasts, hematopoietic stem cells, or plasma cells that produce antibodies. Furthermore, a subgroup of T cells expresses CD20 at lower levels [29]. It is thought that CD20, which is found on the cell surface in tetramers linked to lipid rafts, is involved in the release of calcium from intracellular reserves during B-cell activation. Based on the methods they use to deplete B cells, anti-CD20 mAbs are categorized into type 1 or type 2. Type 1 anti-CD20 antibodies facilitate the cross-linking of CD20, resulting in the aggregation of CD20 molecules within lipid rafts, promoting complement-dependent cytotoxicity (CDC) [30]. Whereas type 2 anti-CD20 antibodies do not activate complement or cross-link CD20 molecules in rafts. Rather, they are more effective than type 1 antibodies at promoting programmed cell death [31]. Every anti-CD20 antibody triggers antibody-dependent cellular cytotoxicity (ADCC) through its interaction with the fragment crystallizable (Fc) domain of the antibody [32]. Anti-CD20 mAbs lead to quick and almost total depletion of B cells in bloodstream, with different speeds of B cell reconstitution [33]. There are three licensed mAbs targeting CD20 available for the treatment of MS patients: Ocrelizumab (Ocrevus^®^, Roche Pharma AG, Grenzach-Wyhlen, Germany), Ofatumumab (Kesimpta^®^, Novartis Pharma GmbH, Nuremberg, Germany) and Ublituximab (Briumvi^®^, Neuraxpharm Pharmaceuticals, S.L., Barcelona, Spain), while Rituximab continues to be used as an off-label alternative. All are type 1 anti-CD20 antibodies.

Both Rituximab and Ocrelizumab are administered intravenously and target very similar and overlapping CD20 epitopes. Rituximab binds to amino-acid residues 168–175 located on the large extracellular loop of CD20 [34], while ocrelizumab focuses on the extensive extracellular loop of CD20 at amino acid positions 165–180 [34]. Ocrelizumab mainly depletes B-cells via ADCC with a lesser role of CDC, whereas Rituximab predominantly triggers higher levels of CDC compared to ADCC [33]. Ofatumumab, administered subcutaneously, attaches to non-contiguous segments of the small (amino-acid residues 74–80) and large (amino-acid residues 145–161) extracellular loops of CD20, sharing an overlap position of the Ublituximab epitope [34]. This epitope is believed to contribute to Ofatumumab’s stronger and more intimate binding to CD20, alongside a slower “off-rate”, compared to Rituximab and Ocrelizumab. It has been suggested that these factors lead to Ofatumumab’s increased CDC in comparison to other antibodies due to more effective binding and placement of complement system components on the target-cell surface [35]. Ublituximab, also administered intravenously, is a chimeric mAb that was developed with low fucose content in its Fc region granting an enhanced affinity to the FcγRIIIa (CD16) receptor on NK cells, and thus increasing the NK-mediated ADCC towards target cells [36]. It targets a unique CD20 epitope comprising two segments of the large extracellular loop, although with partial overlap with each of the other mAbs.

After the depletion phase, which occurs in different times for each anti-CD20 mAb (as illustrated separately for each anti-CD20 agent), a period of replenishment begins. Following Rituximab therapy, immature B cells were observed to reappear first, accompanied by a rise in circulating plasma cells, followed by an increase in the quantity of mature naïve B cells [37,38]. The restoration of memory B cells was gradual and delayed, with levels staying notably lower (<50% of baseline) for over 2 years [37,39].

Comparable results were observed with Ocrelizumab treatment, after which the first repopulating B cell subsets were immature, transitional, and naïve B cells [40]. Additionally, following Rituximab treatment, B cells reappear with a more activated phenotype, evidenced by an increase in CD25, CD40, CD69, and CD86 [39]. Likewise, after Ocrelizumab treatment B cells reappear with an enhanced migratory phenotype, marked by elevated expression of CD49d, CD11a, CD54, and CD162 [40]. Less is known about the effect of Ofatumumab on B cell subsets repletion, nevertheless it is plausible to assume that Ofatumumab could have similar effects of Rituximab and Ocrelizumab [39]. However, it has been proven that time to B-cell repletion with subcutaneous Ofatumumab is faster compared to other anti-CD20 agents, representing a possible advantage in terms of vaccination strategies and reproductive health, alongside treatment switch necessities [41].

Since CD19 is expressed on newly-generated B cells, monitoring CD19+ cells levels in patients undergoing anti-CD20 mAbs treatment offers an extensive and dependable evaluation of B-cell depletion and regeneration [42]. Due to the variability and possible down-regulation of CD20 after treatment, CD19 serves as a more stable marker for monitoring overall B-cell populations. Furthermore, decreased numbers of CD19+ B-cells were linked to lower ARR, EDSS, and fewer Gd+ enhancing lesions, indicating a potential surrogate marker for clinical outcomes [43].

##### Rituximab

Rituximab is an IgG1 murine-human chimeric mAb [34]. It received approval in 1997 for certain types of cancers and autoimmune conditions, including non-Hodgkin lymphoma and rheumatoid arthritis (RA) [44,45]. While it is not officially approved for MS, it is commonly used in an off-label manner, due to its efficacy and relative safety in MS patients as demonstrated through several studies such as the phase I HERMES trial [13] in RRMS and the phase II/III OLYMPUS trial [26] in PPMS showing a marked reduction in total GdE lesions with a relative reduction of 91% [13] compared to placebo, a relative reduction of 61.6% in total T2 lesion volume from baseline [26] and a decrease in relapse rates (14.5% vs. 34.3% at week 24 and 20.3% vs. 40% at week 48) [13].

The HERMES trial showed a rapid and almost complete depletion of CD19+ peripheral B lymphocytes (>95% reduction from baseline) from 2 weeks post-treatment up to 24 weeks after Rituximab treatment. At week 48, CD19+ cells were at 30.7% of their baseline levels [13]. The OLYMPUS trial showed that Rituximab treatment was linked to a quick and almost total (>95%) reduction in CD19+ peripheral B cells from week 2 to week 96 of treatment. At week 122, peripheral B-cell counts were recovered in 35% of Rituximab-treated individuals, where recovery was measured using B cell counts above the lower limit of normal (LLN, 80 cells/μL). Among patients who prematurely ended up the treatment period, 40% of them recovered their peripheral B cells 48 weeks after their final dose [26]. Comparable findings were demonstrated in other studies [46,47], including a study by Bar-Or et al. showing a B-cell depletion reaching approximately 99.8% by week 2 and which was maintained until week 48. B cells were restored to an average of 34.5% of baseline levels by week 72; most of these were naive (CD27-) instead of memory (CD27+) B cells.

##### Ocrelizumab

Ocrelizumab is a humanized IgG1, being the initial anti-CD20 mAb Food and Drug Administration (FDA) authorized in 2017 [27] for MS in the treatment of adults with PPMS and RMS (CIS, RRMS, and active SPMS) [48], according to OPERA I/II [49] and ORATORIO phase III trial [27,50] outcomes. The OPERA I and II trials showed that Ocrelizumab caused a decrease of 94% and 95% in total T1 GdE lesions and resulted in 46% and 47% lower annualized relapse rates (ARR) compared to IFN-β as an active comparator [49]. The ORATORIO study demonstrated that, at 12 weeks, verified disability progression occurred in 32.9% of patients treated with Ocrelizumab compared to 39.3% of those on placebo (*p* = 0.03). At 24 weeks, confirmed progression of disability was observed in 29.6% of patients treated with Ocrelizumab and 35.7% of those in the placebo group (*p* = 0.04) [27].

In the OPERA I/II trials, Ocrelizumab caused a nearly total removal of CD19+ B cells by week 2 following the initial dose, and persisted to be significantly lowered until week 96 [49]. The phase III ORATORIO study in PPMS, which followed the same dosage schedule as the OPERA trials, showed similar results, with nearly total CD19+ B cell elimination from week 2 and held steady for the rest of the 216-week experiment [27].

As regards repletion, in a clinical trial involving 51 patients, the median duration for B-cell counts to revert to either baseline or LLN (80 cells/μL) was 72 weeks (with a range of 27–175 weeks) following the final Ocrelizumab infusion. Within 2.5 years following the final infusion, B-cell counts increased to either baseline or LLN in 90% of patients [51].

##### Ofatumumab

Ofatumumab is presently the sole fully human IgG1 FDA authorized in 2020 for use in all forms of RMS disease (CIS, RRMS, and active SPMS) at low-dose subcutaneous injection [52,53]. In the ASCLEPIOS I and II trials, Ofatumumab reduced GdE T1 lesions by 97% and 94%, and ARR by 50.5% and 58.5%, respectively, when compared to Teriflunomide as an active comparator [54].

The ASCLEPIOS, APLIOS, and APOLITOS trials [54,55,56] indicated quick, efficient, and persistent B-cell depletion after 1–2 weeks. Over 95% of the patients who were evaluated in the phase III ASCLEPIOS studies had B-cell counts below the LLN at every visit from week 2 to week 120 of the research [54]. Similarly, following two weeks of Ofatumumab therapy, B-cell counts below the limit of detection (80 cells/μL) were observed by all patients in the phase II APLIOS study [56]. Furthermore, B-cell depletion to below the LLN was seen for all patients on day 7 of the phase II APOLITOS research [55], and this depletion persisted until week 24.

Following treatment discontinuation, patients treated with Ofatumumab showed faster B-cell repletion rates compared to other anti-CD20 therapies, even though a consistent and complete B-cell depletion is maintained throughout treatment [25,41,57,58]. After stopping subcutaneous Ofatumumab, B cells have been demonstrated to recover over the LLN or baseline, in ≥50% of patients in 24–36 weeks (median 24.6 weeks) [41,59]. This is supported by pharmacokinetic B-cell modelling and simulation, which estimate a median duration of 23 weeks for B-cell replenishment [60], which is lower than that reported for Ocrelizumab and Rituximab. Time to B cell repletion was also influenced by the dosage, appearing to last longer in the higher-dose groups (starting approximately at week 30 for the highest-dosage group and at weeks 12–16 for the lower dosages), with 64–74% of patients achieving the LLN by week 132 [61].

##### Ublituximab

Ublituximab is the most recent chimeric anti-CD20 drug to be FDA approved for intravenous use in relapsing MS (including CIS, RRMS and active SPMS). Because of its enhanced NK-mediated ADCC, Ublituximab is characterized by much faster B-cell depletion compared to other anti-CD20 drugs, achieving an over 96% depletion rate in the first 24 h after infusion [62]. Ublituximab was also linked to an increase in the relative abundance of naïve and regulatory T cells and a decrease in effector memory and central memory T cells, skewing their ratio towards a more anti-inflammatory effect, probably because of its deletion of B APCs [63]. Lastly, Ublituximab succesfully deleted some cytotoxic T cell clones, possibly part of the T subpopulation expressing the CD20 antigen, and temporarily decreased NK cells, probably because of activation induced apoptosis after ADCC [63]. Ublituximab is highly effective in reducing new lesion formation and in achieving the NEDA status: an early phase 2 multicenter trial showed NEDA in 74% of its cohort at 48 weeks, with only 15% of patients showing new lesions [64]. In the ULTIMATE I and II trials, Ublituximab was further proven superior to Teriflunomide in lowering the ARR and the appearance of new lesions on MRI, although without a significant comparative effect on disability worsening, which was low in both groups [62]. In these trials, NEDA was achieved by 44.6% and 43% of patients, respectively [62]. Potential advantages of Ublituximab over other anti-CD20 antibodies encompass its enhanced ADCC capacity, with the potential to more efficiently target low-CD20 expressing cells, the targeting of a different CD20 epitope compared to either Rituximab, Ocrelizumab and Ofatumumab, and its shorter infusion time of 1 h. B cell repopulation dynamics were instead similar to Rituximab and Ofatumumab [35,64].

#### 2.1.2. Adverse Events

##### Hypogammaglobulinemia

Anti-CD20-depleting therapies are associated with a decrease in serum immunoglobulin (Ig) levels, an elevated risk of infections, and a diminished response to vaccinations [65]. Notably, the development of hypogammaglobulinemia can occur months to years following the start of anti-CD20 treatment [66,67], in particular following the third year of treatment [68] and its occurrence becomes more common as treatment continues [69].

Despite the lack of CD20 expression on plasmablasts and plasma cells that produce IgG and IgM, one hypothesis suggests that anti-CD20 agents hinder the B-cell ability to regenerate [67]. Hypogammaglobulinemia may also be explained by the fact that long-lived memory plasma cells, that persist even after B-cell depletion treatment, may not be entirely self-sustaining and might need to be replenished by CD20-expressing B-cell progenitors [70].

Several risk factors for hypogammaglobulinemia have been identified, including older age (≥50 years) [71], lower initial IgG and IgM levels [71,72], treatment duration [73], body mass index (BMI) [73], the use of Rituximab versus Ocrelizumab [71], average annual Rituximab dosage [73]. Prior immunosuppressive treatments such as high-dose chemotherapy, stem cell transplantation [74], cyclophosphamide, mycophenolate mofetil [75,76] and mitoxantrone exposure [73] might be identified as additional risk factors for hypogammaglobulinemia, while the use of other recent prior DMT was not considered a significant risk contributor [71].

A meta-analysis by Elgenidy et al. [77] on IgG levels on over 20,000 MS patients showed an overall occurrence of IgG hypogammaglobulinemia, with defined thresholds ranging from 5–7.37 g/L, in 11% of MS patients who were treated with anti-CD20 mAbs, with differences among drugs: Rituximab had the highest occurrence at 18%, followed by Ocrelizumab at 11%, unspecified anti-CD20 at 10%, and Ofatumumab at 2% [77]. A higher proportion of patients developed hypogammaglobulinemia in a recent multicenter study, with 25% of Rituximab and 18% of Ocrelizumab-treated patients developing hypogammaglobulinemia after a mean of 28 months of treatment [78]. In this study, however, the LLN for IgG levels was 7 g/L.

Data from clinical trials suggest that different anti-CD20 drugs do not impact equally on serum Ig levels [67]. Multiple studies in MS patients have shown decreases in mean IgM and IgG levels and/or a rise in the number of patients with IgM/IgG levels below LLN after receiving multiple cycles of Rituximab [13,68,79] and Ocrelizumab [79,80,81], whereas Ofatumumab resulted in a significant rise in the proportion of patients displaying IgM levels below the LLN, while having minor or no effect on IgG levels [67].

Regarding IgM, the proportion of patients with this Ig isotype below the LLN was 22.4 [13] and 45.8 [82] % for Rituximab, 13.5 [81] and 33.3 [83] % with Ocrelizumab and between 14.3 [84] and 25.1 [85] % for Ofatumumab [67]. Additionally, it has been shown that IgM levels decrease more rapidly than IgG and IgA levels over time, with more patients having IgM levels below LLN as therapy duration increases [67,68,86,87]. Meanwhile, the proportion of patients with IgG below the LLN across clinical trials and real-world evidence studies with Rituximab varied between 3.2 [68] and 28 [82] % and with Ocrelizumab between 1.1 [27] and 13.5 [81] %, while with Ofatumumab the proportion ranged between 1.3 [84] and 1.7% [67,85]. These findings suggest that, compared to Rituximab and Ocrelizumab, Ofatumumab seems to have a lower impact on IgG levels, although the reasons are currently unclear [67]. One hypothesis for the different effect on IgG involves the different binding site of CD20 between Rituximab and Ocrelizumab compared to Ofatumumab, together with the different configurations of these antibodies, which may play a role in various methods of B cell depletion [57], thus impacting IgG- or IgM-producing cells in distinct ways. Moreover, subcutaneously administered antibodies do not deplete splenic B cells, which produce antibodies with higher affinity compared to bone-marrow B cells [88], as seen with intravenously administered antibodies [41], However, it is noteworthy that in clinical trials on Ofatumumab the follow-up period is shorter than in studies conducted on Rituximab and Ocrelizumab and that, unlike the latter two, there are currently few real-world studies available on Ofatumumab.

Ultimately, data on the effect of anti-CD20 treatments on IgA levels are scarce. Some clinical studies revealed that Ocrelizumab had comparable effects on IgA levels as those on IgG [27,80,83]; however, a real-world investigation reported no effects of Rituximab or Ocrelizumab on IgA [79], and extremely limited data indicate modest effects of Ofatumumab on IgA [89].

##### Infections

Clinical relevance of hypogammaglobulinemia during anti-CD20 treatment is linked to the fact that it is one of the risk factors for infections, including severe ones [66] requiring hospitalization, leading to serious consequences even conducing to death [68]. However, IgM type hypogammaglobulinemia, which occurs more frequently and earlier compared to IgG type hypogammaglobulinemia, doesn’t seem to be significantly linked to a heightened risk of severe infections [68]. Several findings suggest in fact a potential connection between susceptibility to infection and decreased levels of IgG, rather than IgM [68,83,90,91,92]. These results emphasize that IgG is the main Ig isotype playing multiple roles in the humoral immune response compared to IgM. Indeed, IgG has the longest serum half-life and the highest concentration in bloodstream and leads to a more effective pathogen clearance, to a strong complement activation and a memory response, all of which are essential in avoiding recurrent and chronic infections. Whilst IgM, with a shorter half-life and a reduced serum concentration, plays a critical role in the early immune response, but it lacks the long-lasting protection and memory associated to IgG [93]. This may perhaps explain why IgG deficiency is more detrimental for immune defence than IgM deficiency.

Research on infection risk with MS treatments discovered that Rituximab had the greatest number of severe infections compared to Natalizumab, Fingolimod, Interferon beta, or Glatiramer acetate with an incidence rate of serious infections of 19.7/1000 person years (PY) [94]. Another real-world study evaluated a severe infection risk of 38.6 per 1000 PY under Rituximab treatment [68]. Furthermore, decreased IgG levels (<6 g/L) at the beginning and throughout treatment was a reliable predictor of severe infections, indicating the need for close monitoring of individuals with low pre-treatment Ig levels [95]. Besides, Rituximab showed higher rates of hypogammaglobulinemia and infections compared to Ocrelizumab [71]. This could be a result of the increased ADCC and decreased CDC observed with humanized Ocrelizumab together with its reduced immunogenic and immunosuppressive risks in contrast to chimeric Rituximab, resulting in a more advantageous safety profile [96].

Regarding Ocrelizumab, according to an integrated safety analysis of phase III trials, the serious infection rate for was 5.68 per 100 PY for patients with IgG below LLN (<565 mg/dL) and 2.16 per 100 PY for patients with IgG > LLN [83]. Similarly to what was observed for Rituximab, data on patients treated with Ocrelizumab showed that decreased serum Ig levels were linked to a higher likelihood of severe infection and the correlation was more robust with IgG levels compared to IgM or IgA levels [67,97]. A more recent pooled post hoc long-term analysis of interventional trials and their open-label extension studies concluded that time on Ocrelizumab and abnormal IgG levels were not significantly associated with an increased severe infection risk, but, rather, abnormal IgM levels. However, authors acknowledge the possibility of attrition bias and a limited generalizability to real-world settings [98].

Concerning Ofatumumab, low incidence of serious infections is reported (<2 per 100 PY) with 5.38% of patients reporting ≥1 severe infection [99], and there was no association observed between a reduction in Ig levels and the risk of serious infections [85,99].

Considering the associations between anti-CD20 mAbs treatment and a heightened risk of infection, concerns regarding the vulnerability of MS patients under treatment and a severe COVID-19 infection were raised [57].

Several studies have in fact demonstrated that anti-CD20 mAbs, Rituximab and Ocrelizumab notably, increase the risk of severe COVID-19 infection and frequent hospitalization compared to other MS treatments, and this may be related to decreased Ig levels [100,101,102,103]. Concerning Ofatumumab, the ALITHIOS study showed that 8.2% of participants experienced COVID-19 infections, with 94.2% classified as mild or moderate and 7.2% as severe [104]. When compared with the overall population, these results indicate that Ofatumumab did not increase patients’ risk of severe COVID-19 infections [104].

Concerning Ublituximab, infections were registered in more than half (55.8%) of the cohort of the ULTIMATE trial, similarly to the pivotal trials of the other mAbs. The most common infection reported was nasopharyngitis (18.3%), although Ublituximab also mildly increased the risk of herpetic infection [62]. Serious infections were observed in 5% of patients in Ublituximab therapy, compared to 2.9% in the Teriflunomide group. Three deaths were registered, with one as the result of pneumonia and one of encephalitis after measles, while the third was because of salpingitis after an ectopic pregnancy [62].

##### Derisking Strategies

Currently, there are no established consensus guidelines for evaluating serum Ig and addressing the risk of hypogammaglobulinemia in MS. The main recommendation in the management of iatrogenic hypogammaglobulinemia is screening and regular monitoring of Ig levels before and during anti-CD20 treatment [105]. Some general strategies for assessing or reducing the risk of hypogammaglobulinemia and the related risk of infections have been suggested [67,105].

Prolonging the interval between two consecutive doses of B-cell-depleting therapy, dose reduction or treatment switch or interruption have been proposed as derisking strategy [106,107], even though further studies are needed to assess how to minimize the chance of a drug’s side effects while maintaining its efficacy. Prophylactic vaccination with inactivated vaccines and immediate antibiotic therapy in a patient under anti-CD20 mAbs treatment developing fever are considered supplementary derisking strategies in patents with secondary hypogammaglobulinemia [108].

Alongside the adjustment or cessation of anti-CD20 treatment, vaccination and antibiotic usage, “immune supplementation” with intravenous Ig (IVIG) administration could be beneficial in patients at high risk of major infection problems [67,105]. Starting IVIG replacement therapy (Ig-RT) is a complicated choice lacking standardized guidelines, and it often requires collaboration between the patient and a multidisciplinary clinical team.

##### Vaccines

Individuals undergoing B-cell-depleting treatments may have a weakened humoral immune responses to vaccines [105,109,110]. A recent systematic review from Vijenthira et al. [111] assessed that vaccination seems to be safe for patients receiving anti-CD20 therapy; however, these patients have a very low response to vaccination, with seroconversion rates ranging from 0% to 25% in research studies while on active treatment. Although the response to vaccination improves gradually over time, it may not reach the same level as healthy controls even after 12 months of therapy [111]. Numerous research studies have assessed vaccination responses in patients treated with Rituximab and revealed diminished humoral responses, varying with the different vaccines [112,113,114].

The VELOCE study assessed responses to specific vaccines in patients with RRMS treated with Ocrelizumab, showing that patients under treatment exhibited reduced humoral responses to the tetanus toxoid vaccine, pneumovax, and the KHL neoantigen vaccine [110].

There are a lack of data regarding the effectiveness of vaccines in patients with MS who are treated with Ofatumumab. However, patients with RRMS treated with Ofatumumab appeared to generate an effective immune response after receiving inactivated influenza vaccination [115], which may be due to quicker B cell repopulation and reduced peripheral depletion after treatment.

Therefore, vaccination must be thoroughly planned for patients undergoing treatment with anti-CD20 mAbs. Live or live-attenuated vaccines can be administered up to 4 weeks before anti-CD20 treatment, while non-live vaccines can be administered up to 4–6 weeks before or at least 3 months after the anti-CD20 treatment [41].

The impact of anti-CD20 mAb treatments on COVID-19 vaccine responses in MS patients is also a focus of interest [57]. Vaccination response in people treated with anti-CD20 mAbs appear to be reduced and different mAbs revealed diverse seroconversion rates after COVID-19 vaccination. In particular, seropositivity after COVID-19 vaccination was assessed in 11%, 43% and 75% patients receiving, respectively, Rituximab, Ocrelizumab and Ofatumumab [116].

In a study assessing patients receiving Rituximab for autoimmune conditions, a diminished antibody response after two doses of the mRNA COVID-19 vaccine was observed compared to healthy controls (29% developed neutralizing antibodies in the Rituximab group versus 92% in the healthy control group) [117].

In another study, merely 25% of patients receiving Ocrelizumab exhibited detectable protective IgG levels 8 weeks after COVID-19 mRNA vaccination, and this response was not sustained at 24 or 36 weeks after vaccination [118]. Nonetheless, a different study revealed a successive rise in the percentage of patients exhibiting an antibody response following each booster dose, and after four booster vaccinations, 90% of patients receiving Ocrelizumab showed an antibody response [119].

Comparable diminished responses to COVID-19 vaccination were noted with Ofatumumab [120,121]. Otherwise, the limited sample size restricts the inferences that can be drawn from this data.

Despite the reduced humoral immune response to SARS-CoV-2 vaccine among patients under anti-CD20 treatment, evidence suggests that T-cell responses might be maintained or even enhanced with anti-CD20 mAb therapy [122], which could alleviate the effects of humoral vaccine response [123]. Currently, the existing expert consensus guidelines recommend administering the COVID-19 vaccine 6 weeks before initiating treatment and at least 3 months after the final infusion [124].

##### Other AEs

Regarding further AEs during the administration of anti-CD20 mAbs, infusion-related reactions (IRRs) must be taken into consideration. They typically manifest within the initial 24 h post-administration, especially after the first dose [57] and they can be reduced by adequate prophylactic treatment. The most probable mechanism for IRRs is a type 2 hypersensitivity reaction accompanied by the release of cytokines [49]. Indeed, the rapid complement activation that occurs after the targeted binding of anti-CD20 mAbs to CD20 molecule on B cells, leads to the formation of different products (like C3a and C5a). Products of complement activation can act as anaphylatoxins and are known to activate macrophages and mast cells, which are significant cytokine sources [125].

Findings from phase I and II studies on Rituximab in MS, along with real world evidence from Zecca et al., showed that the occurrence of IRRs varied from 50.5% to 78.3%, with most of these side effects being mild to moderate in intensity [13,26,126]. In the ORATORIO and OPERA phase III trials, respectively, 39.9% and 34.3% of patients in the Ocrelizumab group experienced IRRs [27,49]. In the ASCLEPIOS phase III trials, 20.2% of patients receiving subcutaneous Ofatumumab experienced injection-related systemic reactions (IRSRs) occurring ≥ 24 h after injection, compared to 15.0% of patients receiving placebo injections alongside oral Teriflunomide [84].

In the ULTIMATE trial, 43.3% of patients developed IRRs during or after the first infusion of Ublituximab, with a successive decrease to under 10% in the second one and further following decline. Nonetheless, only 2.8% of the reactions were severe [62].

A further well-recognized complication related to the immune suppression that goes along with anti-CD20 treatment is the reactivation of latent infections such as tuberculosis, the human immunodeficiency virus and mostly Hepatitis B Virus (HBV) [41]. It is noteworthy that anti-CD20 therapies are frequently utilized alongside other immunosuppressants for conditions beyond MS, including methotrexate for Rheumatoid Arthritis (RA) and polychemotherapy for cancers [127]. HBV reactivation has been observed with Rituximab in the treatment of non-MS conditions like RA, in a patient receiving both Ocrelizumab and methotrexate for RA [128] and in patients receiving Ofatumumab for Chronic Lymphocytic Leukaemia (CLL) [53], albeit at a greater dose but for a shorter time than applied for MS. In clinical trials for MS, HBV reactivation was not observed with Rituximab, Ocrelizumab, Ofatumumab, or Ublituximab [41]. Considering the established risk for HBV flares or reactivation, it is essential to test all patients for HBsAg and anti-HBc infection before starting treatment. Patients showing any positive infection marker (HBsAg+ or HBsAg−/Anti-HBc+) face over a 10% risk of HBV reactivation and should receive antiviral prophylaxis during treatment and for 12 months following therapy discontinuation. Therefore, it is also recommended to conduct a Quantiferon/tuberculosis screening, a chronic hepatitis panel, and check varicella zoster virus (VZV) IgG levels [108].

### 2.2. Alemtuzumab

#### 2.2.1. Mode of Action and Efficacy

Alemtuzumab (Lemtrada^®^) is a humanized monoclonal antibody directed against CD52, a glycoproteic antigen of undefined function expressed on the surface of lymphocytes, monocytes, macrophages, natural killer (NK) cells and monocyte-derived peripheral blood dendritic cells [129] while it is absent on tissue resident dendritic cells, neutrophils and hematopoietic stem cells [130]. Alemtuzumab was formerly known as Campath^®^ and MabCampath^®^, authorized by the US FDA and the European Medicines Agency (EMA) in 2001 for the treatment of B-cell chronic lymphocytic leukaemia (B-CLL) [131]. Afterwards in 2013, Alemtuzumab was approved by the EMA [132] as an effective therapeutic option in patients with RRMS who have demonstrated inadequate response to two or more MS drug therapies, and it is now licensed in over 70 countries.

The efficacy and safety of Alemtuzumab in active RRMS was evaluated through three core clinical trials, which consisted of a 3-year phase II trial (CAMMS223) [133] and two 2-year phase III trials (CARE- MS I) [134] and (CARE-MS II) [135] together with a 5-year follow-up analysis of the abovementioned trials called the TOPAZ study [136]. These studies have shown a high efficacy of Alemtuzumab in MS treatment with a risk reduction on relapse rates ranging between 49.4% [135] and 54.9% [134], a low annualized relapse rate (ARR) of 0.16 [134] and 0.28 [135] and a reduction in disability progression in more than 70% of patients [136], inducing a long-term remission after only two cycles of treatment [134]. Alemtuzumab’s efficacy is due to its role as a potent lymphocyte-depletor leading to extensive and prolonged depletion of both B- and T-cells via ADCC and CDC [137].

During the first month after administration, Alemtuzumab induced rapid and profound lymphopenia with a reduction of over 95% of circulating T and B cells accompanied by less noticeable and temporary effects on monocytes, NK cells, dendritic cells, and neutrophils [138]. The effect on B cells was shorter-lasting than the effect on T cells: B cells returned to original levels after 7 to 10 weeks after Alemtuzumab, even exceeding the pre-treatment level, whereas the recovery of T cells occurred slowly reaching initial levels from 25 weeks [139]. One potential reason for the varying pace of T and B cell repopulation in the blood may be due to the limited impact of Alemtuzumab on the bone marrow. Accordingly, the precursors and early-stage B cells can start to repopulate in a more swiftly manner than that of T cells, which involve thymic participation for complete reconstitution. Additionally, Alemtuzumab causes a partial reduction in single-positive and double-positive thymocytes, which may account for the slower recovery of T cells [139].

Regarding B cells, in individuals with MS, Alemtuzumab significantly reduced peripheral blood B cells (by >85%) within 1 month following every treatment cycle, with cell counts typically returning to baseline levels or surpassing the LLN within 3–6 months after a treatment course [140,141,142,143,144]. In a study assessing immune cell variations in blood weekly during the initial month following the start of treatment, the greatest decline in B cells was observed as soon as 2 days after the completion of the initial treatment course [145].

When analysing B cell phenotypes, the patterns of depletion and repopulation showed differences. Immature B cells quickly expanded (to 160–180% of baseline levels at 3 months), and stayed elevated at 12 months after each treatment cycle [144]. These cells comprised the majority of the B cell population one month post-Alemtuzumab (54% of B cells compared to 7% at baseline), although this prevalence decreased over time as various B cell phenotypes arose (19% at 3 months and 13% at 12 months) [143]. After two years of Alemtuzumab therapy, in fact, predominantly naïve and transitional B cells were identified [144,146]. The maturation into naïve B cells progressed over time, with transitional B cells quickly and nearly entirely diminished at 1 month following the treatment cycle, subsequently returning to baseline levels by 3–6 months before exceeding normal levels (to approximately 130–165% of baseline) [144] and prevailing in the B cell population from month 3 to 12 (constituting roughly 75% of B cells) [143]. Conversely, memory B cells were quickly and nearly entirely diminished at 1 month and stayed depleted by 75% to over 80% at 12 months after the treatment cycle [143,144]. These cells turned into the rarest B cell phenotype as transitional B cells expanded [143]. This prolonged inhibition of memory B cells is believed to play a crucial role in preventing relapses by restricting the resurgence of autoreactive B cells that could drive disease activity [139]. These B-cell modifications have also been associated with changes in the levels of the B-cell activating factor (BAFF) in the blood, which is critical for the survival and development of B lymphocytes [143].

Following the initial treatment with Alemtuzumab, Breg cells notably rose at 5 months and stayed higher until 11 months after the second treatment course [142] and have demonstrated an enhanced ability to generate the anti-inflammatory cytokine IL-10 and effectively prevent the CD4+ effector T cells from proliferating [147]. The regeneration of Breg cells includes both B cells that highly express programmed death ligand-1 (CD19 + PD-L1hi cells), which perform regulatory functions through cell-to-cell contact by interacting with PD-1 on T cells, and the immature transitional B cell subset (CD19 + CD24hiCD38hi) that generates IL-10 [142] Specifically, a lack of CD19 + CD24hiCD38hi B cell subset has been observed in cells during a relapse when compared to both remission and healthy individuals [142,148].

The importance of Breg cells lies in their main role in suppressing the immune system by releasing anti-inflammatory cytokines like IL-10 and by stopping autoreactive T cells and other immune responders from activation, preserving immune tolerance [149]. A lack of Breg function or quantity in MS has been connected to the breakdown of immune tolerance and disease advancement [150].

After receiving Alemtuzumab, the distribution of B cells changes to a more naïve phenotype and the lack of Breg cells is reversed, indicating a potential protective mechanism involving Breg cells [148]. Therefore, it has been proposed that a contributing element to Alemtuzumab’s long-term effectiveness is the sustained decrease in memory B cells and the rise in B cells with regulatory ability [139].

#### 2.2.2. Adverse Events

Despite its high efficacy, the use of Alemtuzumab is limited by the risk of IRRs, opportunistic infections and secondary autoimmune disorders, such as thyroid disorders, immune thrombocytopenia (ITP), and glomerular nephropathies [151].

The most frequent AEs in Alemtuzumab studies were IRRs, which include mild to severe pyrexia, headache, rash, and nausea [152]. In the CARE-MS I-II studies, the incidence of IRRs decreased with subsequent rounds of Alemtuzumab: 85%, 69%, 65%, 63%, and 46%, respectively [152].

##### Autoimmunity

Immune system recovery is responsible for secondary autoimmune diseases observed after Alemtuzumab treatment with the highest occurrence rates between 2- and 3-years post-treatment [153]. In phase III clinical trials [134,154] 29.6% of patients had thyroid disfunctions, while in CAMMS223 [133], were recorded 39% and 29% of patients, respectively, with Alemtuzumab 12 mg and 24 mg. Owing to the prevalence of thyroid disorders, thyroid function tests should be performed both before and every three months throughout treatment [155]. Additionally, a 2% overall incidence of ITP was documented in clinical studies [134,135]. A full blood count is therefore necessary both during the course of therapy and for up to 48 months following the last infusion [155].

The thyroid is the most commonly affected organ by autoimmunity with an occurrence in 17% to 34% of patients [156] with Graves’ disease considered as the primary cause of thyroid dysfunction, accounting for 60–70% of cases [133].

The development of autoimmunity in genetically susceptible individuals may be influenced by several factors, including B-cell depletion, the following hyperpopulation during a phase with lower T cell regulation and the hyperpopulation of naïve B cells in conjunction with a long-lasting depletion of memory B cells [144]. Additionally, T cell recovery results from a peripheral expansion and may support the self-reactive immune cells population [157]. The production of autoantibodies takes months to years after Alemtuzumab treatment due to the need for CD4+ T-cell participation, which recover only 6 months to 3 years after depletion, with a later gap between B cell hyperreactivity and the onset of autoimmunity [144]. Another potential risk factor may be identified in the excessive production of IL-21, which can cause T cells to undergo excessive cycles of growth and cell death, which in turn raises the chances for T cells to come into contact with self-antigens, resulting in the development of autoimmune disease [158,159]. IL-21 also affects B-cell function: B cells’ differentiation into antibody-producing plasma cells depends on IL-21 signalling and CD4+ T-cell cooperation [160]. This could lead to the emergence of antibody-induced autoimmunity [161].

##### Infections

Additional frequent AEs linked to Alemtuzumab treatment are infections with a prevalence rate ranging between 66 and 77% of patients [133,134,135]; in particular, ones from mycetes, herpes zoster, and herpes simplex virus are prevalent and common under Alemtuzumab treatment [139]. Despite the slightly greater incidence of infections in Alemtuzumab groups, these instances are mild to moderate in severity, and they start to decline after the first year of therapy [162]. Severe infections, on the other hand, were considerably less common [133,134,154].

Prophylactic therapy with an oral anti-herpes medication, testing for anti-Varicella zoster virus (VZV) antibodies and vaccination for those who are antibody-negative are necessary as part of infection risk management due to the increased prevalence of herpes virus infections in clinical studies using Alemtuzumab, particularly during the first month after infusion [155]. A heightened risk of human papillomavirus cervicitis (HPV) has been recognized [163] and several cases of Listeria meningitis in MS patients treated with Alemtuzumab 24 mg have also been documented, including an isolated instance of cerebral nocardiosis [162,164,165]. To lower the risk of infection, the suggestions are restricted to antiherpetic prevention, HPV screening and dietary restrictions on food-free Listeria like raw and unpasteurized milk [155].

Concerning the risk of infection during Alemtuzumab treatment, it was primarily hypothesized that patients under treatment may have a greater risk of COVID-19 [166], especially with a severe outcome. However, Iovino et al. [167] reviewed 17 studies related to COVID-19 infection in MS patients undergoing treatment with Alemtuzumab and found that in all examined cases, no severe progression of the disease was recorded, and no deaths were seen. These results were in accordance to those reported by further studies which did not report severe outcomes of COVID-19 disease (no pneumonia, hospitalization, intensive care unit, or death) in people treated with Alemtuzumab [101,168,169].

##### Other AEs

With regard to malignancies, the CAMMS223 study reported a total of 6 malignancies (two papillary thyroid carcinomas, breast cancer, keratoacanthoma, non–small-cell lung cancer, and micropapillary thyroid carcinoma) in patients treated with Alemtuzumab over a 5-year period [133], while the TOPAZ extension study, a 9 years follow-up post-hoc analysis of CARE-MS I and II, showed that three CARE-MS I highly active disease (HAD) patients developed malignancies, whereas none of CARE-MS II HAD patients was affected [170]. Currently, it is unclear if using Alemtuzumab could raise the risk of developing malignant tumours, especially thyroid tumours, as autoimmunity to the thyroid gland could be a contributing factor to this risk [155].

Furthermore, vascular disorders, both cardiac and cerebral, were reported as AEs occurring shortly after infusion [139]. Myocardial ischaemia and infarction, cervicocephalic arterial dissection, cerebral haemorrhage and pulmonary alveolar haemorrhage are now cited as contraindications to the use of Alemtuzumab [171]. It is, therefore, important to perform a baseline electrocardiogram before the beginning of Alemtuzumab infusion [171]. There are 13 reported cases of ischaemic and haemorrhagic stroke and cervical artery dissection after Alemtuzumab usage [172] and 5 cases of spontaneous intracranial haemorrhage [172]. The most reliable hypothesis concerning the underlying mechanism of AEs involves the cytokines release due to immune cells cytolysis which occurs during Alemtuzumab infusion, particularly linked to IRRs as well as for secondary autoimmune diseases [155,173]. Moreover, researchers suggested that hypertension could be the cause of cardiovascular events and this is the reason why intensive screening is essential before starting infusion, especially for patients with a higher blood pressure [172] hence further research are needed to elucidate the causative mechanisms.

##### Vaccines

Although Alemtuzumab causes B lymphopenia, it does not seem to significantly impact immune reactions to vaccines [174]. RRMS patient receiving Alemtuzumab are able to retain immunological memory and respond to a range of vaccines, including those for diphtheria, tetanus, poliomyelitis, Haemophilus influenzae type B, meningococcus C, and pneumococcal polysaccharide [143].

Regarding SARS-CoV-2 vaccination, studies in Alemtuzumab-treated patients have demonstrated an effective response with the development of seroconversion after SARS-CoV2 vaccination [175,176]. The efficient humoral response to the anti-SARS-CoV-2 vaccine was, indeed, similar to that of individuals who were untreated or receiving different DMTs [168]. According to the National Multiple Sclerosis Society [177], for people currently under Alemtuzumab treatment, it is advised to get SARS-CoV-2 vaccination at least 24 weeks following the last administration. For people who are about to start treatment with Alemtuzumab, it is recommended to get fully vaccinated at least 4 weeks before the beginning of the treatment.

### 2.3. Cladribine

#### 2.3.1. Mode of Action and Efficacy

Cladribine (Mavenclad^®^) is a synthetic purine nucleoside analogue (2-chlorodeoxyadenosine, 2-CdA) initially developed in the 1970s as a chemotherapeutic agent for haematological malignancies [178]. In August 2017 Cladribine tablets were approved in Europe for the treatment of adult patients with highly active RRMS and then, in March 2019 in the USA for the treatment of adult patients with RRMS and active SPMS, and many other countries followed suit [179,180].

The efficacy and safety of Cladribine tablets in the treatment of RRMS patients was evaluated mainly in the CLARITY trial [181], a 96-week double-blind placebo-controlled phase III study and its extension [182], in the 2-year randomized, double-blind placebo -controlled ORACLE-MS study [183], and the ONWARD trial, a 2-year randomized double blind phase IIb study [184]. In the CLARITY study, Cladribine tablet treatment led to a substantial decrease in the ARR, with a relative reduction of 55–58% [181].

Its primary mode of action is the selective depletion of lymphocytes, especially B and T cells. After administration, specific nucleoside transporter proteins allow cells to uptake Cladribine, which is then phosphorylated by deoxycytidine kinase (DCK) to provide the mononucleotide 2-chlorodeoxyadenosine 5′-monophosphate (2-CdAMP). The active molecule 2-chlorodeoxyadenosine 5′-triphosphate (2-CdATP) is produced by further phosphorylation processes [178]. As cells divide, 2-CdATP enters their DNA, causing DNA strand breaks and, eventually, apoptosis [185]. The majority of cells contain 5′-nucleotidases (5′-NTase), which counteract DCK activity and prevent the production of 2-CdATP [178]. Although further experimental evidence is required to prove this, the degree of phosphorylation at Ser74 appears to control DCK activity [186]. Compared to other cell types, phosphorylation occurs more frequently in B and T lymphocytes because they have consistently high levels of DCK and relatively low levels of 5′-NTase [178]. The elevated DCK level is believed to play a crucial role in lymphocyte clonal expansion in both development and immune responses [187]. Therefore, they are particularly susceptible to the accumulation of 2-CdA nucleotides [178]. Results from DCK messenger RNA profiling studies indicate that whereas DCK levels and the ratio of DCK to 5′-NTase are relatively low in many non-hematologic cell types, they are elevated in T cells (CD4+ and CD8+), B cells, and dendritic cells. The adverse event profile of Cladribine may be positively impacted by non-hematologic cells having a decreased susceptibility to the drug [188,189].

Evidence indicates that Cladribine tablets function as an oral short-course immune reconstitution therapy (IRT) [190]. This approach leads to a transient decrease in lymphocytes with a predominance in B-cell and T-cell populations, followed by gradual reconstitution that happens at varying times and rates [140,191,192,193,194].

Compared to the extremely quick decreases observed following treatment with mAbs using a cytolytic mode of action, lymphocyte reductions following Cladribine treatment are quite moderate [27,140]. Combined findings from CLARITY, CLARITY extension, and PREMIERE reveal that during the lymphocyte depletion phase following the initial treatment cycle, B cell counts declined by 70% at week 5, 81–84% at nadir (13 weeks), approximately 60% at week 24, and roughly 30% at week 48 [195]. These decreases occurred more rapidly and were more significant than those seen in T cells. In fact, the latter decreased by roughly 50% by week 5, mainly due to CD4+ T cell depletion, while median CD8+ T cell levels remained above the reference range, indicating that this subset is relatively resistant to the lymphodepleting impacts of Cladribine [195].

Every treatment cycle is succeeded by a gradual repopulation of lymphocytes. B cells returned to levels within the reference range by week 84, approximately 30 weeks post the final treatment dose, while the recovery of median CD4+ T cell counts to normal levels was more gradual, taking up to 96 weeks [194].

These effects on lymphocytes are expected to play a key part in Cladribine’s therapeutic benefits in MS patients [194], while the exact mechanisms by which Cladribine works remain partly unknown. Variable timing and kinetics of B cell subset reduction and reconstitution were observed.

Several studies [191,192,193,196,197] indicated that, in contrast to anti-CD20 mAbs, which specifically target CD20+ B cells, Cladribine causes a broader removal of B cell types, particularly evident in the memory compartment, yet there were no alterations in peripheral Ig levels [193,198,199].

Particularly, CD19+, CD20+, memory, naïve, and activated (CD69+) B cells decreased early in the treatment, indicated by profound median percentage changes from baseline to month 1 [193] with nadir occurring at month 2. While CD19+, CD20+ and CD69+ B cells demonstrated reconstitution toward baseline levels from month 3, naïve B cells began to recover by month 2 and approached baseline levels by month 12, with a complete recovery resulting in a mild hyper-repopulation at the end of year 1 [191,193,196,197,200]. Studies demonstrate that naïve B cells recover more rapidly than memory B cells, causing the immune system to become more naïve and less autoreactive [181,193,196].

Regarding memory B cells, encompassing both unswitched (IgD + CD27+) and class-switched memory B-cells (IgDCD27+), they underwent a significant and sustained depletion and remained reduced until month 12 [191,193,196,200,201]. Indeed, the remaining memory B cell clones were fairly large in size and clonally expanded [197]. While memory B cells expressing only IgM (CD19 + CD27 + IgD- IgM+) and CD19 + CD27 + IgD-IgM- class-switched memory B cells continued to be notably depleted, unswitched memory B cells (CD19 + CD27 + IgD+) reconstituted faster and the treatment-induced decreases were no longer significant by month 24 [191].

The marked vulnerability experienced by memory B cells compared to other B cell subsets may be associated with B cells’ high DCK to 5′-NTase expression ratio, notably in mature, memory, and germinal centre B cells, but not in plasma cells [202]. It has been proposed that the significant and enduring reduction in memory B-cells is one of the ways Cladribine achieves prolonged effectiveness following complete lymphocyte recovery [203].

Plasmablasts were slightly affected by Cladribine treatment [192,193,196,197,200]. The reduction in plasmablasts occurred more gradually and after that of naïve and memory B cells, reaching the nadir at month 3 [193], attaining peak depletion at week 8 following the initial cycle of Cladribine and at week 60 after the subsequent cycle [193,200]. The recovery of plasmablasts was delayed and took place over a prolonged period as demonstrated by their decreased levels throughout time even after the first year of treatment [192,193,196,200].

Plasma cells, which are terminally differentiated and non-dividing cells, have demonstrated reduced vulnerability to Cladribine [192,198,199], in line with the observation that they exhibit a relatively lower expression ratio of DCK to 5′-NTase in contrast to the notably high levels found in other B cell subtypes [202]. Week 6 represented an exception with a sudden decline in plasma cell counts. Subsequently, plasma cells rose to peak levels during phase when other B cells and plasmablasts exhibited their largest reduction [192].

Breg and B transitional (Btrans) cells were reduced at month 1 in a comparatively quick but less significant manner. Btrans cells start recovering at week 14 while Breg cells started to repopulate early in month 2 but did not fully decrease after the second cycle of Cladribine [192,193]. By month 3, cell counts for both Breg and Btrans cell subtypes had recovered and exceeded baseline levels [193]. Breg and Btrans cell counts continued to rise beyond baseline levels until month 12, perhaps increasing the ratio of regulatory versus effector B-cell subtypes from months 2 and 3 [193]. The early recovery of Breg cells, which produce anti-inflammatory cytokines such as IL-10, helped re-establishing normal B cell balance [191,192]. These results could be connected to the immune system’s effort to reestablish a balanced immunological setting and ultimately inhibit autoreactive lymphocytes [191].

The selective depletion followed by rebuilding of the immune system with different impact on B cell subsets observed after Cladribine treatment, is crucial in regulating the immune response and maintaining long-lasting control of the disease in MS, reinforcing the idea that Cladribine functions through immune reconstitution instead of prolonged immunosuppression.

#### 2.3.2. Adverse Events

As outlined in the following sections, Cladribine could be considered a safe treatment option with few well-known or uncommon side effects [204].

Headache was the most frequent treatment-emergent adverse event (TEAE), with rates of 8.71 per 100 PY for Cladribine tablets 3.5 mg/kg and 8.82 per 100 PY for placebo [205].

Hematologic AEs were more frequent with Cladribine tablets and included lymphopenia, leukopenia, and neutropenia, in decreasing occurrence [205]. Lymphopenia is due to Cladribine’s mode of action, which leads to selective and transient lymphocyte depletion. In clinical trials, lymphopenia was reported more frequently as an AE for Cladribine tablets 3.5 mg/kg groups than for placebo ones (adjusted AE incidences per 100 PY 7.94 vs. 1.06 for placebo) [205]. Approximately 25% of patients treated with Cladribine tablets at a dose of 3.5 mg/kg in clinical trials developed grade 3 lymphopenia (ALC < 0.5 × 10^9^ cells/L) throughout the two-year course of therapy, while less than 1% of patients suffered grade 4 lymphopenia (ALC < 0.2 × 10^9^ cells/L) at any point during the same period [205].

##### Infections

Although Cladribine’s capacity to specifically deplete and reconstitute lymphocytes, contributes to its effectiveness, the period of lymphopenia that goes along with it raises worries regarding the risk of infection [206]. In CLARITY trial infections or infestations occurred in 47.7% of patients in the Cladribine 3.5-mg group, 48.9% of those in the Cladribine 5.25-mg group, and 42.5% of participants in the placebo group. The majority of events were assessed as mild or moderate by the investigators and the frequency of infections in the Cladribine groups had an inverse connection with the lowest lymphocyte count [207]. The most common infections seen with Cladribine tablets, occurring at a rate of at least double that of placebo, were viral upper respiratory tract infection (3.0 vs. 1.1%), vaginal infection (1.9 vs. 0.2%), and herpes zoster (1.9 vs. 0%) [181]; particularly, patients with grade 3 or 4 lymphopenia were found to have the highest incidence of herpes zoster infections and upper respiratory tract infections [207]. Herpes zoster was a notable adverse event linked to Cladribine tablets in MS studies over two years [179] with an incidence of 0.83 per 100 PY compared to 0.20 per 100 PY for placebo and it was more common during grade 3 or 4 lymphopenia (2.16 vs. 0.75 per 100 PY) [205]. All cases were dermatomal and progressed normally, without post-herpetic neuralgia, only one was serious [205]. Nevertheless, as a safety measure, patients who are seronegative because of lacking previous exposure to VZV, should receive vaccination before starting Cladribine tablets, and treatment initiation should be delayed for 4–6 weeks [179].

However, serious infections were reported in 2.9–2.3% of patients in the Cladribine groups compared to 1.6% in the placebo one [207]. The fact that serious infections are not common in Cladribine treated patients possibly may be due to the unique dosing timetable that enables immunological recovery intervals between treatment cycles [194].

As regards COVID-19, available data concerning patients treated with Cladribine tablets indicated that they were not typically at any higher risk of significant illness and/or a severe outcome with COVID-19 compared to the general population and other people with MS who acquired COVID-19 [208,209].

##### Malignancies

A further consideration to evaluate when using Cladribine is the possibility of an elevated risk of malignancies. Concerns regarding a potential link between Cladribine and a higher risk of cancer were expressed by early research [207].

However, the evaluation of clinical trial and long-term safety information in a conclusive report from the clinical development program indicated that there was no significant rise in the rate of cancers with Cladribine tablets in comparison to placebo; the occurrence of malignancy was 0.26 per 100 PY in the treatment group versus 0.12 per 100 PY in the placebo group [205,210]. When this group of patients was compared to an external reference population, the cancer rates were similar for those treated with Cladribine tablets and the matched cohort [205,210,211]. Moreover, there were no clustering of specific malignancies, and no rise in malignancies usually linked to weakened immune systems was found (such as blood disorders, virus-triggered, or non-melanoma skin cancers) [205,210].

##### Vaccines

A substudy of the MAGNIFY-MS cohort has shown that patients treated with Cladribine tablets maintained sufficient immunity from seasonal influenza and VZV vaccines up to 6 months after Cladribine treatment [212]. The CLOCK-MS substudy assessed how prior treatment with Cladribine tablets could affect the formation of antibody titers in response to influenza vaccination [213]. Three out of four individuals included in the vaccine substudy had protective antibody titers against seasonal influenza 4 weeks after immunization. Two of these individuals exhibited lymphopenia around the date of vaccination and had received treatment with Cladribine tablets up to 4 months before the vaccination [213]. Likewise, Cladribine does not seem to impact preexisting antibody levels to prevalent pathogens [214].

Current recommendations [180] advise that Cladribine tablet treatment must not begin within the 4- to 6-week window following vaccination with live or attenuated vaccines because of the potential risk of active vaccine infection. Live or attenuated live vaccines should be avoided during and after the use of Cladribine tablets as long as the patient’s white blood cell counts remain outside normal limits. Lymphocyte levels need to be observed until they normalize or reach at least >800 cells/mm^3^ [190]. Patients who are seronegative for the VZV should be vaccinated before starting Cladribine treatment [180].

The immune responses to SARS-CoV-2 vaccination were not affected in patients treated with Cladribine [215], and seropositivity after vaccination was not dependent on lymphocyte counts or age [216]. Crucially, antibodies levels remained stable for 6 months after vaccination [217]. Likewise, Cladribine does not seem to impact preexisting antibody levels to prevalent pathogens [214].

## 3. Sequestering Therapies

Sequestering therapies seek to reduce intrathecal inflammation by peripherally segregating lymphocytes, whether in blood or in lymphoid organs, thus impeding their crossing of the blood brain barrier. As they do not cause lymphocytic depletion, they are generally associated with a lesser infectious risk when compared to other HETs, but they can still alter immune functionality and have the downside of possible rebound MS activity upon discontinuation. Presently used sequestering therapies include Natalizumab and sphingosine-1-phosphate receptor modulators.

### 3.1. Natalizumab

#### 3.1.1. Mode of Action and Efficacy

Natalizumab (Tysabri^®^) is a humanized monoclonal antibody of the IgG4 class approved for the treatment of RRMS following the AFFIRM trial [218]. It was first designed to prevent the entrance of pathogenic T lymphocytes into the CNS by blocking the α4 subunit (also known as CD49d) of integrin α4β1 (or Very Late Antigen-4, VLA-4), a surface molecule that mediates lymphomonocytic extravasation through CNS endothelium by interacting with vascular cell adhesion molecule 1 (VCAM-1) [219]. Since its approval, Natalizumab demonstrated its short-term high efficacy in multiple studies [220], while its long term benefit was proven by the large and real world Tysabri Observation Program (TOP) study, which highlighted a higher than 90% reduction in ARR and a significant reduction in disability worsening, with a cumulative probability of remaining relapse free and stable in EDSS for 10 years of 45.8% and 72.2%, respectively [221].

Evidence of Natalizumab action on B lymphocytes emerged soon after approval, as Niino et al. first demonstrated a higher expression of the target molecule VLA-4 on B lymphocytes compared to T cells, also demonstrating that Natalizumab effectively blocked their in vitro migratory capacity through brain endothelial cells [222]. Consistent with this data, multiple subsequent flow cytometry studies in patients treated with Natalizumab showed a significant and sustained increase in the absolute and relative numbers of circulating lymphocytes, with CD19+ B cells increasing significantly more than their T cell counterpart, underscoring their greater sensitivity to the drug [223,224,225,226,227,228,229,230]. Particularly, although a truly precise comparison is hindered by heterogenous study populations and different flow-cytometry gating definitions, most of the studies agree that Natalizumab increases circulating mature memory subsets, richer in VLA-4 expression [231], with an increase in CD19+ CD27+ IgD- memory B cells consistently documented [225,229,230,232,233,234]. Moreover, other demonstrated a peripheral enrichment in CXCR3+ B cells, which are normally characterized by an enhanced CNS migratory potential in MS [227,235].

This increase in peripheral blood mature B cells can be partly attributed to Natalizumab’s titular mechanism of action, as various flow cytometric studies on the CSF demonstrated a concomitant significant reduction in CSF leukocytes, and particularly of CD19+ memory B cells, CD138+ plasma cells and plasmablasts [236,237,238,239]. Moreover, studies on functional parameters of intrathecal B activity such as IgM and IgG indexes and OCBs reinforce this idea, as they were found to be generally decreased, although not completely abolished. In particular, the IgG index was consistently shown to be reduced by different papers [225,238,239,240,241], noting a greater decrease in stable patients [239] and a longitudinal inverse correlation between intrathecal IgG levels and time in treatment [241]. Results on the IgM index are less concordant, but mostly show a decrease in CSF IgM levels, possibly more significant in patients without evidence of disease activity [238,239,241]. Lastly, earlier reports suggested a complete or nearly complete OCB disappearance in patients without disease activity [239,242]. Nonetheless, subsequent studies confirmed this finding only in a minority of the total patient population, while adding that OCBs were observed to become weaker in an additional portion of patients [238,240,241,243]. The decrease in CSF immunoglobulins during Natalizumab therapy was further confirmed by a recent proteomic study, which also showed a broader shift towards an anti-inflammatory and reparatory CSF milieu, although without pointing out other frankly B cell specific processes beyond the decrease in intrathecal antibody production [244]. Taken together, these findings suggest that Natalizumab is effective in quantitatively reducing intrathecal B cell related inflammation, although it is less effective on its qualitative aspect and generally does not completely abolish antibody production. As the magnitude of IgG intrathecal synthesis is thought to be dependent on plasmablasts, and as they are the B lineage subtype richest in VLA-4 [241], these data point out their effective blockage, while at the same time suggesting the persistence of plasma cells, responsible for OCB persistence, in their CNS survival niches [241]. Interestingly, the weakening of OCBs seen during Natalizumab therapy is not present in patients treated with anti-CD20 antibodies, suggesting that this effect is possibly mediated, apart from the decrease in CSF plasmablasts, also by an impairment of immune cross-talk as T cells too are prevented from entrance to the CNS [241,243].

Peripheral blood flow cytometry studies further suggest that the specific action on B lineage cells of Natalizumab is not limited to blocking their CNS extravasation but has far-reaching VLA-4-dependent effects on their adhesion to primary and secondary lymphoid organ niches and on their activity and survival. In fact As VLA-4 has been shown to be critical in B cell retention in the splenial marginal zone [245], and as the spleen is the greatest reservoir of these B subtypes, Natalizumab has been suggested to cause their release [233,246].

Regarding this, data about immature B cell phenotypes changes during Natalizumab are more conflicting; overall, flow cytometric studies seem to mostly agree on an absolute increase in immature B cell phenotypes during Natalizumab therapy [223,230,232,233,246,247,248]. Consistent with this, the investigation of kappa deleting recombination excision circles (KRECs) and T cell receptor excision circles (TRECs), a reliable estimate of newly produced T and B lymphocytes, has revealed their increase [246]. These data, together with the demonstration of an increased egress of CD34+ bone marrow lymphoid-committed progenitors by Planas et al. and by Zohren et al. [233,249], has suggested that Natalizumab causes a premature egress of immature cells from lymphoid primary organs.

Nonetheless, the increase in immature B cell subtypes, possibly associated with regulatory properties [232], is rendered less relevant by the much larger increase in mature B lymphocytes, that causes an overall B cell population to shift towards a pro-inflammatory milieu. In fact, one study [229] demonstrated an increased production of pro-inflammatory cytokines and of costimulatory molecules during Natalizumab therapy, while also showing a direct in vitro pro-inflammatory effect of Natalizumab stimulation. This is potentially explained by the presence of a mild direct signaling effect of CD49d upon its binding [250]. In addition, Natalizumab has also been shown to alter miRNA regulation [251,252]. As Natalizumab is greatly effective in preventing intrathecal inflammation [220] and, as noted before, has an anti-inflammatory intrathecal effect [244], its pro-inflammatory effect on peripheral blood B cells testifies on the effectiveness of its blockage of CNS extravasation, while at the same time it could explain the possible rebound disease activity reported after its discontinuation [250].

#### 3.1.2. Adverse Events

##### Infections

While confirming Natalizumab’s great efficacy, the TOP study has also expanded the insight into its safety, confirming past clinical trials [220] and showing that infections were the most common adverse events, with an overall incidence of 4.1% [221]. Among these, Progressive Multifocal Leukoencephalopathy (PML), a form of encephalitis caused by the John Cunningham virus (JCV) had an overall incidence of 0.9% in the TOP study, and was the cause of half of the 8 registered deaths (0.13%) [221]. Although rare, because of its gravity, PML is overall considered the main and most important risk associated with Natalizumab therapy; thus, close clinical, neuroradiological and serological monitoring has become the standard of care to reduce it as much as possible [220]. A portion of the peculiar risk of PML during Natalizumab can be explained by its impairment on CNS T cell surveillance, with reports suggestive of a CD4+/CD8+ ratio inversion in the CSF [253], as cellular immunity is thought to be more important than the humoral response in JCV control [254]. The importance of CNS immunosurveillance is further underscored by the significant over 85% reduction in the risk of PML when adopting an extended-interval dosing for Natalizumab administration, allowing partial VLA-4 functionality with minimal CNS immune repopulation, while at the same time controlling MS activity [255]. Nonetheless, the unique increase in risk of PML during Natalizumab therapy in comparison to other drugs suggests that other mechanisms are at play beyond immunosuppression; in particular, the significant mobilization of B cells from primary and secondary lymphoid organs during Natalizumab therapy could hold some explanatory power. As briefly mentioned above, PML is caused by the lytic infection of oligodendrocytes by JCV, a polyomavirus that is usually acquired in early life and then stays quiescent in the kidneys, but also in the spleen and bone marrow tissues [254]. JCV has in fact been repeatedly shown [256,257,258] to infect B cells and hematopoietic progenitors, although one study failed to confirm this last finding [259]. Thus, Natalizumab could mobilize great numbers of potential JCV-carrying cells into circulation; however, how these could penetrate and infect the CNS is still not clarified. As the drug was found to not completely saturate VLA-4 receptors in vivo, it could be hypothesized that blockage in CNS migration induced by the drug is not complete [222,237]. Additionally, Natalizumab was shown to be capable of enhancing JCV gene expression through an increase in the transcription factor Spi-B, capable of binding JCV noncoding control region in the presence of specific viral mutations that can occur, albeit with high inter-patient variability, during infection [260,261]. The association between Spi-B and JCV was further strengthened by another study, that found a significant increase in this factor in peripheral mononuclear blood cells only in JCV+ patients. Moreover, POU2AF1, a transcriptional coactivator for Spi-B, was found to be increased too in JCV+ patients, but decreased in JCV latent patients compared to patients with JCV DNA detectable in blood and urine [262].

Other reported infections in the TOP study were mostly mild or moderate, with an increased risk of pneumonia (0.4% of the cohort), mild UTIs (0.3%) and herpetic infections (0.3%) [221]. An increased risk of SARS-CoV-2 infection, although without an accompanying rise in COVID severity, was also highlighted by another study [263]. Conversely, Natalizumab does not seem to increase the risk of malignancies compared to the general population [220,221]. Lastly, other significant adverse events registered in the TOP study were hypersensitivity reactions (0.7%) and hepatic events (0.2%), both mostly mild or moderate [221].

##### Vaccines

Lastly, Natalizumab has been shown to alter the percentage of circulating plasmablasts. While an earlier study adopting the definition of CD19+ CD138+ cells for plasmablasts observed their increase during Natalizumab therapy [223], two subsequent studies, using different definitions (CD19+ CD27++ CD38++ [264] and CD3- CD20- CD27++ [265]), have found the opposite, with a reduction in frequency and an unaltered absolute value. As it has been shown how VLA-4 is critical for plasmablast development in germinal centers and subsequent survival [266,267], and also acts in B cell bone marrow homing and as a facilitator of B cell activation [264,268], the contemporary reduction in both systemic and CSF plasmablasts suggests that Natalizumab impairs their survival and proliferation. Consistent with this, multiple studies showed a decrease in blood IgG and IgM levels during Natalizumab therapy, with one [269] specifying a state of IgG hypogammaglobulinemia in 19.6% of patients, while the same was not observed for IgA, which are VLA-4 independent in their generation [240,264,269,270,271]. The impaired B cell activation during Natalizumab therapy is also consistent with the demonstration of the unique metabolic signature, with higher rates of quiescence, evident in antigen-trained B cells during therapy [272]. Nonetheless, while a trend towards an inadequate response to immunization in Natalizumab patients could be identified, most studies did not find it significantly different from that of untreated patients [273,274,275,276,277].

### 3.2. Sphingosine-1-Phosphate Receptor Modulators

#### 3.2.1. Mode of Action and Efficacy

Drugs of the sphingosine-1-phosphate (S1P) receptor modulator class (Fingolimod, Ponesimod, Siponimod, Ozanimod) are a group of small molecules characterized by functional antagonism on the S1P receptor family. All of them have been approved for use in RRMS and active SPMS by the FDA, while the EMA has approved Fingolimod, Ponesimod and Ozanimod for RRMS only, and Siponimod for active SPMS only [278]. The S1P receptor family encompasses five receptors (S1P1, S1P2, S1P3, S1P4 and S1P5), each with different tissue tropism. In particular, S1P1, S1P2 and S1P3 are ubiquitous, while S1P4 is expressed in lymphoid tissue and S1P5 in NK cells and white matter cells, primarily oligodendrocytes [279]. Most importantly for the mechanism of action of S1P receptor modulators, S1P1 is expressed on lymphoid cells and mediates their egress from primary and secondary lymphoid organs. In particular, S1P1 is upregulated inside lymph nodes, thus promoting lymphocytic migration along the S1P gradient that increases from lymphoid tissue towards blood and lymph; moreover, S1P1 probes the S1P molecules expressed by the cortical sinusoid endothelium, directly mediating lymphocyte passage through it [280]. In fact, S1P1 downregulation was observed to empty cortical sinusoids and to trap most lymphocytes inside secondary lymphoid organs [279,281]. This mechanism was found to be the basis for the immunosuppressive effect of FTY720, the first S1P receptor modulator to be marketed, later known as Fingolimod (Gilenya^®^). Fingolimod, phosphorylated into its metabolite (S)-FTY720-P, acts as a functional antagonist to S1P1, S1P3, S1P4 and S1P5, with higher affinity for S1P1 and S1P4, causing their surface downregulation and intracellular degradation, with subsequent lymphocyte sequestration inside lymph nodes and thymus. This, in turn, drives a dose-dependent peripheral lymphopenia with consequent reduced T cell CNS infiltration and therapeutic effect in EAE [279,282]. The therapeutic effect of S1P receptor modulators in MS was later shown to also have more complex roots in S1P1-mediated amelioration of BBB dysfunction and in S1P1 and S1P5-mediated direct action on CNS cellular survival and myelination [279,283]. Nonetheless, their effectiveness in humans is still thought to be mainly dependent on lymphocytic sequestration, with a preferential action on T cells, and particularly on CD4+ T helper cells, which Fingolimod rapidly reduces to 10% of their baseline value [284]. This effect results in a high-intermediate efficacy against MS, with the pivotal FREEDOMS trial for Fingolimod showing a 54% reduction in ARR and a 30% decrease in disability progression compared to placebo [285]. Other S1P receptor modulators besides Fingolimod were developed to have better receptor selectivity on S1P1 and S1P5, thus hoping to reduce side effects dependent on off-target mechanisms. Similarly to Fingolimod, Siponimod (Mayzent^®^), Ponesimod (Ponvory^®^) and Ozanimod (Zeposia^®^) were shown to cause a reduction in total blood lymphocyte count, mainly dependent on CD4+ T cells, and particularly on CCR7+ T cells [286,287,288]. Consistent with this, their pivotal trials showed similar beneficial effects compared to Fingolimod [278], although Siponimod was additionally shown to reduce 3 and 6 month disability progression in SPMS patients by 21% and 26%, respectively, compared to placebo [289].

Although the action of S1P receptor modulators is preferentially explicated on T cells, soon after the first proposal of Fingolimod, different studies found evidence of a significant retention effect on B cells too, with consequent reduction in B cell CNS infiltration [290,291].

Early evidence from animal studies proved that B cells were prevented from lymph node egress through cortical lymphatic sinusoids after Fingolimod administration; provoking changes in lymph node cytoarchitecture, with moderate increase in follicle B cell density and B cell displacement in cortical T cell areas [281]. Subsequent flow cytometry studies in humans further demonstrated an entrapment action on B cells, showing that Fingolimod caused an absolute decrease in circulating B lineage cells [237]. Different B cell subtypes were found to be differentially affected by Fingolimod in later research. In particular, mature memory B cells were univocally found to be decreased both in absolute values and percentage [272,292,293,294,295,296,297,298,299]. Plasmablasts were found to be decreased in absolute values, with a substantially unchanged relative presence in two studies [292,293], while one showed an increase in their relative population [297]. The literature is more discordant on naïve B cells [292,293,294,295,297,298,300,301,302], while other earlier B populations, such as immature B cells, usually defined as CD19+ CD21- cells, and transitional B cells, were mainly found to be increased in their relative quantity [247,292,297,300,303]. Lastly, regulatory B cells, broadly defined as CD38+ CD24+ CD27- cells, were found to have only little absolute decrease with a significant relative increase in the majority of studies [295,296,298,299,301,302]. Overall, Fingolimod was shown to increase the circulating proportion of earlier B cell types, together with regulatory B cells. This finding was strengthened by a study finding a relative increase in KRECs in Fingolimod treated patients [300]. Moreover, a transcriptome profiling study showed an increase in genes preferentially expressed by transitional B cells [304]. The expansion of transitional and regulatory B cells was linked to increased IL-10 production and a lower TNF-α production, with an overall anti-inflammatory effect [292,295,305]. This was further strengthened by other studies, which showed an increase in anti-inflammatory TGFβ+ and IL-4+ B cells, as well as BTLA+ B cells, involved in the inhibition of immune responses [305,306]. The mechanism underlying the differential lymphocyte sequestration during Fingolimod therapy is not entirely known, but some authors proposed that a different profile in the expression of chemokine receptors could partly explain it, for example showing that naïve B cells were higher in CCR7 expression, while lowest on L-selectin, highly expressed in memory subtypes [295]. Another study found a preferential targeting of CXCR4+ B cells [306]. Also, Fingolimod was found to increase circulating BAFF, while at the same time decreasing BAFFR+ receptor levels, with these data correlating to the level of CD19+ CD27- CD10+ CD38hi CD24hi transitional B cells, known to be dependent on BAFFR for their maturation [303,307]. Finally, a decrease in costimulatory molecules was noted, potentially explaining the significant decrease in mature B cells [292,294]. Taken together, these findings provide evidence that Fingolimod fosters a B cell-mediated anti-inflammatory effect. This was confirmed by a transcriptome profiling study, that showed a decrease in pro-inflammatory cytokines gene expression while showing an increase in anti-inflammatory ones [308]. The same paper also showed that the mechanism of Fingolimod goes beyond lymphocyte retention, as it modified genetic expression of inflammatory mediators downstream of the B cell receptor (BCR) path and altered master regulators of different lymphocytic pathways and of NF-kB and Wnt/β-catenin pathways [308].

In-depth studies of the specific effect on B cells of selective S1P receptor modulators are lacking in comparison to Fingolimod. Nonetheless, one paper found that, contrariwise to most studies on Fingolimod, Siponimod caused a decrease in naïve B cells frequency in secondary progressive MS patients, while it did not alter CD27+ memory B cells percentages [309]. Nonetheless, an increase in transitional and regulatory B cells, defined as CD24hi CD38hi, was found, along with CD43+ CD27+ B1 enrichment, with an overall shift in IL-10 equilibrium and gene regulation, including that of costimulatory molecules, antigen-receptor signaling and cytokine interaction, in favor of anti-inflammatory processes [309]. On the other hand, another study on secondary progressive patients found an increase in naïve regulatory B cells, while observing a decrease in memory regulatory B cells and total B lymphocytes in secondary progressive patients [310]. Uniquely, this study linked disability progression with a lack of drug effect on B lymphocytes and on CD3+ CD20+ T cells, with the latter shown to be reduced in responders to therapy [310].

Lastly, no effect of Fingolimod was found on leptomeningeal enhancement [311], in accordance with animal models, in which the formation of ectopic meningeal follicles was shown to be prevented by pre-treatment with the drug before EAE onset only [312]. Siponimod was shown to prevent the formation of ectopic meningeal follicles if administered before EAE onset too, but if given at the peak of disease, a decrease in CD220+ B cell infiltration in murine CNS was still observable, as well as a decrease in meningeal follicles in size and number [313].

#### 3.2.2. Adverse Events

##### Infections

The sequestration mechanism, as a downside, is responsible for peripheral lymphopenia, the main adverse effect of S1P modulators. This was found to be present and significant in the majority of Fingolimod-treated patients, although its magnitude was shown to be more dependent on variable CD8+ T cell levels rather than on B and CD4+ T helper cells, which showed a more stable and constant decrease [314]. A similar effect was seen during Siponimod therapy in a SPMS cohort, with 35% and 53% of patients developing grade 2 and 3 lymphopenia, respectively [309]. Ponesimod showed similar lymphopenia rates to the aforementioned drugs, while Ozanimod showed a lower lymphocyte reduction compared to other S1P modulators (57% versus 70%) [278]. Nonetheless, possibly because S1P receptor modulators do not destroy lymphocytes, and thus preserve some of their functionality while they are sequestered, different studies could not demonstrate a correlation between total lymphocyte levels and an increased risk of infections [284,314,315,316]. A higher infection risk was instead associated with neutropenia, a rarer adverse effect [315].

Although not strictly dependent on the grade of lymphopenia, a small increase in the risk of infection during S1P therapy was still detected as a potential adverse effect of S1P modulators [278,317]. In particular, Fingolimod was shown to increase the risk of upper respiratory tract, urinary tract and herpetic infections (respective risk ratios: 1.22; 1.41 and 1.77 [318]; respective incidences in the FREEDOMS trial: 49.9%, 8% and 8.7%) [284,285,316,317,318]. Moreover, sparse reports of cryptococcal infections during Fingolimod therapy are present in the literature [319,320,321], as well as a few reports of PML, although with much lower relative risk compared to Natalizumab, not warranting specific preventive action [322]. SARS-CoV-2 infection risk also may be more elevated [323]. Nonetheless, most COVID cases observed during Fingolimod therapy were mild, and the drug also showed the potential to attenuate possible pathologic hyper-inflammation responses [263,324,325]. Overall, the increased risk of these viral and atypical infections is generally thought to be more reflective of impaired T cell, rather than B cell, surveillance [315,319,322,326].

##### Other AEs

A deficit in cellular immune surveillance could also explain the small increase in the risk of skin cancer, particularly basal cell carcinoma (2% of the cohort in the FREEDOMS trial), during Fingolimod therapy, warranting adequate dermatological surveillance [278]. An S1P1- and S1P3-mediated cardiovascular effect is another possibly significant adverse event associated with S1P receptor modulators, leading to transient heart rate reduction at therapy initiation, with possible bradycardia (0.6% of the FREEDOMS cohort), hypertension (8%) and rare atrioventricular blocks (first-grade 4.7%; second-grade 4%) [278]. Moreover, a very small increase in macular edema (0.5% of the FREEDOMS cohort), likely because of altered vascular permeability, was noted upon therapy initiation [278,317]. Other possible side effects include hepatotoxicity, with a transient Alanine aminotransferase increase of over 3 times the upper normal limit in 14% of patients in the FREEDOMS trial, and a small decrease in functional respiratory measures, mostly with little clinical significance [278].

As already mentioned, selective S1P modulators were designed to reduce adverse events, generally succeeding, especially in reducing cardiovascular effects [278,327,328].

##### Vaccines

Even if some level of lymphocyte functionality appears to be preserved during S1P receptor modulators therapy, Fingolimod was found to alter the response to vaccination, with evidence of lower seroconversion rates against influenza vaccine compared to placebo in a randomized study [329]. Moreover, different papers demonstrated decreased seroconversion, IgG levels and percentages of patients achieving neutralizing titers after SARS-CoV-2 vaccination during Fingolimod therapy [277,330,331], with two studies finding an inverse correlation between vaccine response and time on treatment [277,330]. Nonetheless, booster vaccinations against SARS-CoV-2 were found to be useful in increasing seroconversion rates and IgG titers [277,332]. As Fingolimod was found to alter the development of high-affinity class-switched antibodies while having no influence on IgM production, as well as on the reaction against T-independent antigens in mice [333], the suboptimal responses could be explained by a reduced B-T interaction [277]. In addition, further B specific effects could be hypothesized, as Fingolimod has been shown to alter lymph node cytoarchitecture [334] and to displace splenic marginal-zone B cells to splenic follicles [334,335], thus potentially affecting B cell contact with antigens and their subsequent antigen handling [334]. Less data is available on the vaccination response during selective S1P receptor modulators therapy. Two studies found a better responses against SARS-CoV-2 vaccination compared to Fingolimod, with Ozanimod patients developing a comparable response to an untreated cohort [331,336]. Nonetheless, a third study contradicted these findings, showing a similar overall response to the booster dose in Fingolimod patients and selective S1P modulators, although in the context of an overall satisfactory antibody production in over 85% of patients [332].

## 4. Future Perspectives

Efforts in finding novel therapeutic approaches to MS are ongoing. Bruton’s Tyrosine kinase inhibitors, granting a novel and unique B-specific mechanism of action, have entered phase 3 study in MS and could possibly soon enter clinical practice. In addition, a brief paragraph will be dedicated to possible future variations on the high-efficacy and well-known anti-CD20 agent Ocrelizumab.

### 4.1. Bruton’s Tyrosine Kinase Inhibitors

Bruton’s tyrosine kinase (BTK) is a non-receptor kinase, member of the TEC kinase family. It is found in different hematopoietic cells, including B lymphocytes and myeloid cells, whilst not being present in T cells [337]. In B lymphocytes, BTK is activated as a part of the downstream pathway of B cell receptor (BCR) signaling, serving to trigger different pathways of functional activation and effector function through PLCγ2 phosphorylation and, among others, subsequent PI3K, MAPK and NF-κB signaling [337]. This increases pro-inflammatory cytokine and antibody production, as well as B–T cell interaction [337,338]. In myeloid cells, BTK is activated downstream of FcγR activation, as well as being involved in the degranulation of mast cells and basophils through FcεR and in neutrophil recruitment, thus promoting an overall pro-inflammatory response [337]. While BTK is fundamental in the early life and genesis of B lymphocytes, as its absence results in X-linked agammaglobulinemia (XLA), characterized by the practical absence of cells of the B lineage [337], it is not necessary for mature B cell survival [339]. Thus, these data, together with the fact that BTK over-expression was linked to autoimmunity in mice and human patients [338], suggest a potential for therapeutic BTK inhibition in B and myeloid cell driven autoimmune and oncological disease [337]. While the first BTK inhibitors (BTKi) were characterized by suboptimal selectivity and had significant adverse events dependent on off-target effects, limiting their use to the oncological field [340], newer and significantly selective molecules [341] have been developed and been confirmed to dampen aberrant B cell function without a depleting effect in animal models of autoimmune disease and early human trials [337,338,342,343]. In particular, one study showed that BTKi determined a decrease in B cell mitochondrial respiration, with consequent reduced expression of B cell costimulatory molecules both in the resting and activation states and anti-inflammatory shift in B cell response [338].

The demonstration of enhanced baseline BTK activity in MS patients’ B cells, with poorer inducibility, together with the recent general appreciation of B dependent pathogenic mechanisms in the disease, made the potential for BTK targeting in MS promising [344]. Moreover, the finding of an increased BTK expression in active and chronic active lesions, and particularly in the microglia, opened to the possibility to act on some of the mechanisms underlying the progressive aspect of MS [345]. The study of Evobrutinib, one of the first selective BTKi, in the EAE animal model showed promise in ameliorating symptoms, and was found to correlate with a reduction in B cell infiltrates in the CNS, in encephalitogenic T activation by antigen presenting B cells and in the expression of costimulatory molecules MHC II and CD86 [346]. Furthermore, Evobrutinib impaired the transition from follicular II to follicular I B cells, suggesting an inhibitory effect on the differentiation of B cells into activated phenotypes [346]. Further studies confirmed a reduced B cell infiltration in the inflamed meninges of EAE animals in response to Evobrutinib, with a corresponding decrease in MRI meningeal enhancement, as well as a reduction in astrocytes near inflammatory plaques [347]. Evobrutinib was also shown to reduce the in vitro T-bet and CXCR3 expression of B cells, molecules associated with a pronounced CNS infiltrating ability and tendency to mature into antibody secreting cells [344]. In a B cell dependent EAE HuMOG model, Remibrutinib, another BTKi, was shown to inhibit neuroinflammation successfully [348]. Regarding progressive disease, Ibrutinib, an older and non-selective BTKi, decreased EAE severity in a Biozzi mice model thought to better represent secondary progressive MS [349].

Nonetheless, while Evobrutinib showed effectiveness in reducing new MRI GdE lesions [343] and Slowly Expanding Lesions (SELs) volume [350] in phase 2 trials, it also proved ineffective in reducing the ARR [343]. This was confirmed in a successive randomized control trial (RCT), in which Evobrutinib failed to meet the primary endpoint of reducing ARR compared to teriflunomide in RMS [351]. Notably though, the ARR in Teriflunomide patients was reported to be lower than the known average. Moreover, complete results are still awaiting publication, and it remains to be verified if the drug could influence progression. In addition, it should be noted that the ARR may not represent an adequate measurement tool, as it is increasingly recognized that most of the disability in both RRMS and PMS is accrued independently of relapses [351,352]; moreover, a decreasing trend in ARR has been recently observed in MS, even when considering placebo-treated patients [353]. As inflammatory disease activity can already be satisfactorily controlled by existing drugs, while progression independent of relapses still eludes treatment, the use of outcomes that are better representatives of the mechanisms underlying the latter, such as composite disability progression measurements, neurophysiologic outcomes, and MRI and fluid biomarkers, should be considered, while focusing on ARR could limit potential breakthroughs [351,354].

Findings similar to those of Evobrutinib are to be published regarding Tolebrutinib, another selective BTKi, which proved effective in in reducing new GdE and T2 lesions in a phase 2b RCT and had a better CNS penetration compared to Evobrutinib, although without showing an effect on SELs [345,355]. In particular, in the HERCULES phase 3 RCT in SPMS, Tolebrutinib was found to delay disability progression compared to placebo, while the GEMINI trials did not find it to improve ARR compared to teriflunomide, although confirming a delayed onset of disability worsening and a reduced disability accrual [356]. Interestingly, preliminary data presented at ECTRIMS 2024 showed Tolebrutinib decreased disability progression compared to Teriflunomide, notwithstanding higher radiological inflammatory activity [357].

Regarding adverse events, Evobrutinib and Tolebrutinib showed a generally good safety profile, with the most commonly observed adverse events being headache, nasopharyngitis, UTIs and increases in pancreatic and hepatic enzymes [343,358]. Concerning the latter, preliminary data from the GEMINI trials has highlighted a safety signal, with an early increase in alanine aminotransferase (ALT) higher than 3 times the upper normal limit in 5.6% of the treated cohort, while an increase over 20 times was seen in 0.5%. ALT successively normalized without sequelae in all patients. Evobrutinib was also shown to cause a decrease in IgM, but not IgG [343], and to preserve the immune response against influenza and SARS-CoV-2 vaccines [359]. Other selective BTKi, such as Fenebrutinib, Remibrutinib, Orelabrutinib and BIIB091, are currently in the early stage of phase 3 trial study [360].

Overall, the early data suggest a limited capacity of BTK inhibitors in controlling MS inflammatory activity, as defined by relapses and GdE lesions. Nonetheless, some clues on the efficacy of these drugs on the progressive aspect of the disease have emerged, and deserve additional investigation, using more appropriate outcomes, since options for progressive MS without inflammation are scarce.

### 4.2. Future Advancements on Ocrelizumab

Progressive MS is currently the form of disease with less treatment options available, with one being Ocrelizumab, shown to have a partial effectiveness in reducing disability and MRI progression in PPMS [27]. Nonetheless, Ocrelizumab does not pass the blood brain barrier, and thus cannot largely affect the local smoldering inflammation that is hypothesized to drive disease progression independent of relapse activity [361]. To overcome this limitation, antibodies capable of binding the transferrin receptor-1 protein on the BBB and use it to penetrate the CNS have been developed with the registration of the Brainshuttle™ trademark. The application of the Brainshuttle™ technology to anti-CD20 antibodies is currently under investigation in a phase Ib study [362], and it holds the potential to be a fascinating development in MS care.

Another way to possibly impact disease progression with Ocrelizumab could be by maximizing its exposure, which was in fact correlated with a better B cell depletion and better disability progression outcomes in both RMS and PPMS, although to a lesser extent in the latter [363]. As a simple way of increasing exposure, Ocrelizumab is thus being tested in its effectiveness and safety at a higher dosage regimen (1200 mg or 1800 mg every 6 months, based on patient weight, as opposed to 600 mg every six months) in two complementary RCTs in RRMS and PPMS patients, expected to reach primary completion in December 2024 and April 2025, respectively [364,365]. The primary outcome will be the impact on the time to 12-week confirmed disease progression.

## 5. Conclusions

B lymphocytes have an important role in the pathogenesis of MS. Accordingly, this review highlights how all HETs for this disease were found to have a specific and unique impact on B lymphocyte numbers/proportions (Figure 1), or even functionality, and how research focused on B cells has yielded some innovative possible future therapeutic options, such as BTKi. The distinctive effect of each HET on different B cell sub-populations contributes to its particular benefits and long-lasting efficacy, but it also partly shines light on its adverse event profiles (summarized in Table 1). Additionally, the overall effects of HETs on B cell subtypes, summarized in Table 2, can guide therapeutic switches, alongside knowledge on differing B-cell replenishment rates. Indeed, this review also highlights the differences between anti-CD20 agents, including their repopulation dynamics and, possibly, their varying hypogammaglobulinemia risks, and analyzes possible de-risking strategies, while underlining the importance of monitoring IgG levels before and during treatment. In summary, an in-depth knowledge of the immunological mechanisms underlying each HET, comprising the B cell compartment, contributes to the personalization of treatment by informing the risk-benefit balance assessment and treatment sequencing, and by guiding de-risking and vaccination strategies.

## Figures and Tables

**Figure 1 cells-14-00606-f001:**
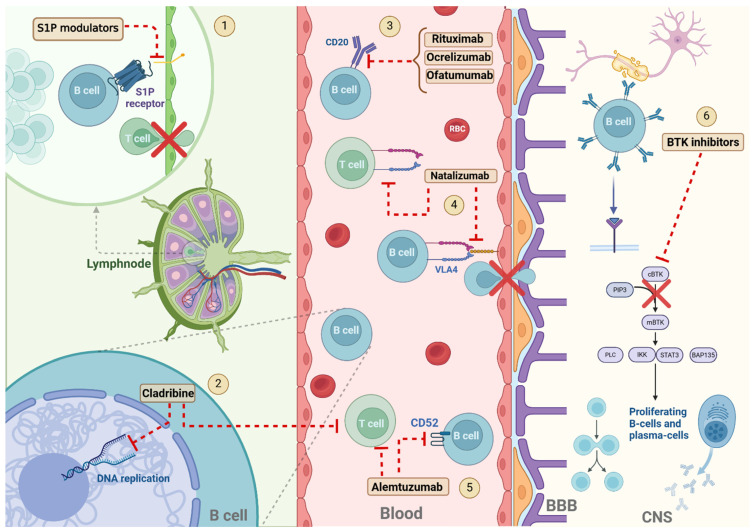
Mechanisms of action of HE-DMTs. 1. Scheme 1. Phoshate (S1P)-modulators bind to the S1P1 receptor, which regulates the egress of B and T lymphocytes from lymphoid tissue. This causes the internalization of the receptor, inhibiting lymphocytic migration outside of lymph nodes. 2. While cells divide, the active compound of Cladribine, 2-CdATP, integrates into their genetic material resulting in DNA strand fractures and ultimately, cell death. 3. Anti-CD20 monoclonal antibodies bind the CD20 antigen expressed on the lymphocyte surface, leading to B cell depletion through ADCC and CDC. 4. Natalizumab binds the α4β1 integrin (VLA-4) expressed by B and T lymphocytes, preventing its binding to the VCAM expressed by Blood-Brain Barrier (BBB) endothelium, thus blocking lymphocytic extravasation to the inflamed Central Nervous System (CNS). 5. Alemtuzumab binds to the CD52 expressed on lymphocytes and other cellular types, triggering the ADCC and CDC mediated extensive and prolonged depletion of both B- and T-cells. 6. Bruton’s Tyrosine Kinase inhibitors (BTKi) prevent lymphocytic activation mediated by the B cell receptor (BCR) by blocking the key enzyme of its downstream intracellular activation pathway, Bruton’s Tyrosine kinase. PIP3: phosphatidylinositol-3,4,5,-trisphosphate; cBTK: Cytosolic BTK; mBTK: membrane associated BTK; PLC: phospholipase C; IKK: inhibitor of NF-κB kinase; STAT3: Signal transducer and activator of transcription 3; BAP-135: 135 kDa BTK-associated protein. Created in BioRender. Galota, F. (2024) https://BioRender.com/b20x457 (accessed on 9 April 2025).

**Table 1 cells-14-00606-t001:** Contribution of B-cell depletion/modulation/sequestration to drug-related adverse events.

	Drugs				
Adverse Events	Anti-CD20	Alemtuzumab	Cladribine	Natalizumab	S1P-Modulators
Lymphopenia	+	+	+	NA	+
Hypogammaglobulinemia	++	NA	NA	++	NA
Reduced vaccine effectiveness	++	+/−	+/−	+/−	+
Infections other than PML	++	+	+	+/−	+
PML	+	+	NA	+	+/−
Neoplasms	NA	NA	+/−	NA	+/−
Cardiovascular effects	+/−	+	NA	NA	−
Macular oedema	NA	NA	NA	NA	−
Hepatic toxicity	−	−	−	−	−
Infusion reactions	+	+	NA	+	NA
Autoimmunity	+	++	NA	NA	NA

Key: ++ mostly related to B-cell depletion/modulation/sequestration; + in part related to B-cell depletion/modulation/sequestration; +/− insufficient/discordant literature data; − unrelated to B-cell depletion/modulation/sequestration; NA not applicable.

**Table 2 cells-14-00606-t002:** Impact of HE-DMTs on the relative proportions (%) of B lineage cell types.

Blineage Cell Types Drugs	Pre-B	Immature	Transitional	Naïve	Memory	Regulatory	Plasmablasts	Plasma Cells
Anti-CD20 mAbs	↓	↓	↓	↓↓	↓↓	-↓	↔	↔
Alemtuzumab	↓	↓	↓	↓↓	↓↓	↓	↓	↓
Cladribine	-	-	↓	↓	↓↓	↓	↓↓	↓
Natalizumab	↑	↑	↑	↑↓	↑↑	-	↓	-
S1P-modulators	-	↑	↑	↑↓	↓↓	↑↑	↔	-

Key: ↑ relative increase; ↓ relative decrease; ↑↑ high relative increase; ↓↓ high relative decrease; ↑↓ discordant literature, prevalent increase; ↔ no relevant effect, concordant literature; -↓ discordant literature, prevalent no relevant effect; - absence of literature. The table summarizes data obtained using partially discordant literature gating definitions, the different phenotypes were mostly defined as follows: Pre-B cells: CD19+ CD10+; Immature B cells: CD19+, CD21-; Transitional B cells: CD19+, CD27-, CD24hi, CD38hi; Naïve B cells: CD19+ CD21+ IgD+ CD27-; Memory B cells: CD19+, CD27+ IgD-/+; Plasmablasts: CD19+ CD27++ CD38++; Regulatory B cells: CD38+, CD24+, CD27-/+ (also CD5+ and IL-10+ in some studies); Plasma cells: CD138+.

## Data Availability

Data sharing is not applicable.

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
