# Peer review of "Impact of High-Efficacy Therapies for Multiple Sclerosis on B Cells"

_cells, 2025, doi:10.3390/cells14080606_

Round 1

Reviewer 1 Report

Comments and Suggestions for Authors

The authors prepared a very detailed review on the influence of DMTS on B cells. The topic of the study is definitely interesting, and potentially of current interest both for clinicians and researchers.

However, the manuscript in present form is too long and in great part is based on simple repeating data from consecutive studies. More focus should be laid on drawing conclusions and summarizing or organizing the knowledge available in the literature.

There are also multiple other issues which have to be improved:

The provided definition of Multiple sclerosis is rather “uncommon” and does not correspond to the citation used. Based on the current knowledge, the autoimmune side should be balanced with neurodegenerative processes.

OCB are not present in the majority but not all of MS patients

Efficacy in “slowing progression in Primary Progressive (PP)” MS was shown for Ocrelizumab only.

Antigen CD20 is not expressed on cells of the B cell lineage at “ALL stages of development”,

Ublituximab belongs to the group of antiCD20 therapies already approved in MS, relevant literature is easy and commonly accessible. Thus Ublituximab should not be mentioned as a future perspective.

The authors should explain the role of CD19 in the monitoring of B cell depletion.

What is a “standard dose of Rituximab” in MS?

The authors should indicate clearly which statements relate to SM and which are based on other patients populations. As an example – the unclear meaning of the sentence “Furthermore, after reconstitution, B-regulatory (Breg) cells, which are immune-regulatory cells that produce TGFβ, IL-10 and IL-35 to support immunologic tolerance and prevent disease recurrence [42,43], may be present in the population of naïve B cells [44].” Recurrence of which disease? What does it mean “may be present”?

The references should be checked for correct descriptions e.g. Ref 41, 68, 69

The particular anti-CD20 DMTs differ in many aspects, not only the level of humanization, but also Ab molecule modifications, target epitopes, the mode of cytotoxicity, administration route. There is no such general information in the text.

The authors mention the results of OPERA studies but omit the very important and unique clicnial results of  ORATORIO study.

For Ofatumumab the authors provide the FDA based approval ” in all forms of RRMS disease and active Secondary Progressive MS”. Similar indications in RMS are listed by FDA also for Ocrelizumab (and Ublituximab) – “Relapsing forms of multiple sclerosis (MS), to include clinically isolated syndrome, relapsing-remitting disease, and active secondary progressive disease, in adults”. The authors should present data for particular DMTs in an uniform way.

The statement “Ofatumumab has been shown to CAUSE faster B-cell repletion compared to other anti-CD20 therapies” is rather unfortunate – the anti-CD20 Ab cannot cause B-cell repletion…

The authors completely ignored the influence of presented DMTs on disease progression, which in fact is nowadays the main point of clinical interest.

In contrast to Rituximab, there is completely no information about the influence of Ocrelizumab and Ofatumumab on B cell subsets, although Ocrelizumab /Ofatumumab are the registered DMTs.

The reference 101 is insufficiently described, and there are more recent data/publications in this regard which allow for different conclusions.

The authors use for explanation of anti-CD20 Ab differences in infection risk a statement “ADCC might be, indeed, more efficient in extravascular tissues and could act even with reduced levels of antigen expression and availability compared to CDC” based on Ref. 100. However this statement was used to explain the differences in the mechanism of action and IRR but NOT in the risk of infections.

Anti-CD20 therapies are commonly associated with the increased risk of reactivation of viral hepatitis which is not mentioned in the manuscript. Similarly, reactivation of other latent infections (except for JCV) as a therapy risk is not addressed in the review.

The clinical meaning of the herpes zoster risk in patients treated with cladribine in the periods of high grade lymphopenia should be better emphasized.

The indication for Siponimod registered by EMA is “active” SPMS. As stated earlier the authors should precisely describe the formal indication of particular DMTs. The sentence “All of them have been approved for use in RRMS and active SPMS by the FDA, while only Siponimod has been approved for use in SPMS by the EMA [291]” does not mention “active” SPMS and suggests that Siponimod is indicated in Europe also for RRMS, which is not correct.

What does it mean that “Ozanimod performed slightly better” in the context of lymphopenia?

Comments on the Quality of English Language

Some sentences are very complicated and hard to follow - the language could be improved.

Author Response

We thank the reviewers for their time and effort. We carefully considered their comments and tried to address all raised issues. We think the review is greatly improved thanks to the suggested revisions.

1) The authors prepared a very detailed review on the influence of DMTS on B cells. The topic of the study is definitely interesting, and potentially of current interest both for clinicians and researchers.

However, the manuscript in present form is too long and in great part is based on simple repeating data from consecutive studies. More focus should be laid on drawing conclusions and summarizing or organizing the knowledge available in the literature.

Response: we thank the reviewer for pointing out the excessive length of our paper and its improvable focus on drawing conclusions from the literature. We agree that it is indeed an extensive work, to the detriment of its readability. To address this issue, and to make room for the suggested additions, we shortened or eliminated some paragraphs that we found least informative and/or least related to the topic of the review. In particular, we shortened/eliminated:

  • guidelines on the start of IgG-replacement therapy;
  • specifications that were beyond necessary on Cladribine efficacy, effect on T cells, adverse events and vaccines;
  • conflicting data about the effect of Natalizumab on immature B cell types
  • conflicting data about naïve B cells during treatment withS1P modulators
  • conflicting data about the impact of S1P modulators on the CSF compartment
  • the excessively lengthy introduction on future perspectives

Other smaller cuts and summarizations were made throughout the whole text to better focus on drawing conclusions from the available literature and improving English readability on the anti-inflammatory aspects of Natalizumab and on its effect on vaccinations, as well as on the utility of booster vaccinations during treatment with S1P modulators. We furthermore hope that the included figure and tables contribute to providing a general overview on the drugs’ mechanisms of action and adverse events mediated by B cells.

2) The provided definition of Multiple sclerosis is rather “uncommon” and does not correspond to the citation used. Based on the current knowledge, the autoimmune side should be balanced with neurodegenerative processes.

Response: the provided definition of MS has been changed as follows: “Multiple Sclerosis (MS) is defined as a chronic inflammatory and neurodegenerative autoimmune disease that affects the Central Nervous System (CNS), characterized by demyelination with axonal loss, astroglial proliferation and grey matter impairment in genetically susceptible hosts

3) OCB are not present in the majority but not all of MS patients

Response: we specified that OCB are present in the majority of MS patients (“Introduction” section -Section 1).

4) Efficacy in “slowing progression in Primary Progressive (PP)” MS was shown for Ocrelizumab only.

Response: we clarified that Ocrelizumab is the sole approved treatment with demonstrated efficacy in PPMS forms (Section 2.1.1).

5) Antigen CD20 is not expressed on cells of the B cell lineage at “ALL stages of development”,

Response: the erroneous wording was rephrased as follows: “Anti-CD20 antibodies specifically target the CD20 (“Cluster of differentiation 20”) molecule, which is expressed on pre-B cells in the bone marrow, as well as on naïve, memory B cells and early plasmablasts in the lymphoid tissues or germinal centers”

6) Ublituximab belongs to the group of antiCD20 therapies already approved in MS, relevant literature is easy and commonly accessible. Thus Ublituximab should not be mentioned as a future perspective.

Response: the section on Ublituximab has rightly been moved from the “future perspectives” paragraph to the currently approved therapies, as well as modified to present it uniformly with other anti-CD20 mAbs.

7) The authors should explain the role of CD19 in the monitoring of B cell depletion.

Response: we explained the role of CD19 as a marker for monitoring B-cell repopulation (Section 2.1.1).

8) What is a “standard dose of Rituximab” in MS?

            Response: as correctly pointed out, the standard regimen for rituximab was not indicated. Due to the length of the manuscript, and in accordance with the type of information provided for the other anti-CD20 agents, information on dosing regimens has not been added and any reference to a “standard” dose of rituximab has been eliminated.

9) The authors should indicate clearly which statements relate to SM and which are based on other patients populations. As an example – the unclear meaning of the sentence “Furthermore, after reconstitution, B-regulatory (Breg) cells, which are immune-regulatory cells that produce TGFβ, IL-10 and IL-35 to support immunologic tolerance and prevent disease recurrence [42,43], may be present in the population of naïve B cells [44].” Recurrence of which disease? What does it mean “may be present”?

Response: this sentence referred to the evidence that Breg cells increase in numbers during the inflammatory phase of several autoimmune disorders (ref 42,43). We aimed to highlight the re-emergence of Breg cells following Rituximab therapy. This section was rephrased (Section 2.1.1) to present the information regarding various anti-CD20 mAbs in a more uniform manner.

10) The references should be checked for correct descriptions e.g. Ref 41, 68, 69

Response: we corrected the specified ones and verified the remaining ones, correcting them, when inaccurate. Thank you very much for prompting this re-examination.

11) The particular anti-CD20 DMTs differ in many aspects, not only the level of humanization, but also Ab molecule modifications, target epitopes, the mode of cytotoxicity, administration route. There is no such general information in the text.

Response: these differences are, indeed, relevant. We accordingly expanded the differences between anti-CD20 mAbs in terms of administration routes, mode of cytotoxicity, Ab molecule modifications and target epitope (Section 2.1.1).

12) The authors mention the results of OPERA studies but omit the very important and unique clicnial results of ORATORIO study.

Response: we appreciate the suggestion to emphasize the distinct clinical outcomes of the ORATORIO study. We therefore added information on the decrease in disability progression, demonstrated in the Ocrelizumab group compared to placebo (Section 2.1.1, subsection Ocrelizumab).

13) For Ofatumumab the authors provide the FDA based approval ” in all forms of RRMS disease and active Secondary Progressive MS”. Similar indications in RMS are listed by FDA also for Ocrelizumab (and Ublituximab) – “Relapsing forms of multiple sclerosis (MS), to include clinically isolated syndrome, relapsing-remitting disease, and active secondary progressive disease, in adults”. The authors should present data for particular DMTs in an uniform way.

Response: we presented indications for the approved anti-CD20 in a more uniform way (Section 2.1.1; subsections Ocrelizumab, Ofatumumab, Ublituximab).

14) The statement “Ofatumumab has been shown to CAUSE faster B-cell repletion compared to other anti-CD20 therapies” is rather unfortunate – the anti-CD20 Ab cannot cause B-cell repletion…

Response: the term “cause” is, indeed, infortunate. The sentence was rephrased as follows: “patients treated with Ofatumumab showed faster B-cell repletion rates compared to other anti-CD20 therapies”.

15) The authors completely ignored the influence of presented DMTs on disease progression, which in fact is nowadays the main point of clinical interest.

Response: we agree that the impact of therapies on disease progression is, indeed, one of the main points of clinical interest. To better represent this aspect, we outlined the findings of ORATORIO (section 2.1.1, subsection Ocrelizumab), while the effect of Siponimod on disease progression was specified in section 3.2.1 (“Siponimod was additionally shown to reduce 3 and 6-month disability progression in SPMS patients by 21% and 26%, respectively, compared to placebo”). We would like to point out that some data on the effectiveness of anti-CD20 mAbs and Siponimod on progression were presented in the “Mode of action and efficacy” paragraphs of each of these drugs (section 2.1.1 and 3.2.1). Moreover, part of the BTKi section is centered on this important issue, which is, to-date, an unmet need.

16) In contrast to Rituximab, there is completely no information about the influence of Ocrelizumab and Ofatumumab on B cell subsets, although Ocrelizumab /Ofatumumab are the registered DMTs.

Response: information on the impact of anti-CD20 agents on B-cell subsets and on the replenishment phase of all agents (including a comment on the scarcity of data concerning the impact of ofatumumab on B-cell subsets) is now featured in the paragraph on the mode of action of anti-CD20 agents (Section 2.1.1) and has been eliminated from the paragraphs concerning the single anti-CD20 agents.

17) The reference 101 is insufficiently described, and there are more recent data/publications in this regard which allow for different conclusions.

Response: we thank the reviewer for underlining the need for more up-to-date data/publications on serum immunoglobulin levels and the risk of severe infections. We mentioned results from an oral presentation of ECTRIMS 2019. We updated the reference by including the more recent systematic review article of Alvarez et al. (Ref 70) which confirmed a greater association between infection risk and IgG hypogammaglobulinemia rather than with IgA and IgM.

18) The authors use for explanation of anti-CD20 Ab differences in infection risk a statement “ADCC might be, indeed, more efficient in extravascular tissues and could act even with reduced levels of antigen expression and availability compared to CDC” based on Ref. 100. However this statement was used to explain the differences in the mechanism of action and IRR but NOT in the risk of infections.

Response: this attempt at proposing a potential mechanism for the varying risk of infections associated with anti-CD20 mAb was removed (Section 2.1.2, subsection Infections).

19) Anti-CD20 therapies are commonly associated with the increased risk of reactivation of viral hepatitis which is not mentioned in the manuscript. Similarly, reactivation of other latent infections (except for JCV) as a therapy risk is not addressed in the review.

Response: we added clarifications on the risk of viral hepatitis reactivation as well as of other latent infections such as tuberculosis in section 2.1.2, “other AEs” subsection, together with suggested recommendations.

20) The clinical meaning of the herpes zoster risk in patients treated with cladribine in the periods of high grade lymphopenia should be better emphasized.

Response: we better emphasized the risk of herpes zoster during high grade lymphopenia in Cladribine treated patients (section 2.3.2, subsection “Infections”).

21) The indication for Siponimod registered by EMA is “active” SPMS. As stated earlier the authors should precisely describe the formal indication of particular DMTs. The sentence “All of them have been approved for use in RRMS and active SPMS by the FDA, while only Siponimod has been approved for use in SPMS by the EMA [291]” does not mention “active” SPMS and suggests that Siponimod is indicated in Europe also for RRMS, which is not correct.

Response: indeed, the indications for Siponimod were incorrectly described and the sentence was unclear. We rephrased it as follows: “All of them have been approved for use in RRMS and active SPMS by the FDA, while the EMA has approved Fingolimod, Ponesimod and Ozanimod for RRMS only, and Siponimod for active SPMS only”.

22) What does it mean that “Ozanimod performed slightly better” in the context of lymphopenia?

Response:  we specified percentages on the comparative lymphocyte drop from baseline between Ozanimod and other S1P modulators, reformulating the sentence as follows “Ponesimod showed similar lymphopenia rates to the aforementioned drugs, while Ozanimod showed a lower lymphocyte reduction compared to other S1P modulators (57% versus 70%).

Reviewer 2 Report

Comments and Suggestions for Authors

You provide a comprehensive and extensive (this latter observation is discussed with the Editor), review of this practical and actuality topic of global interest. I have a couple of minor commentaries on your manuscript. 

In my opinion, a comment should be made on the Association of British Neurologists Guidelines expressed in 2015, regarding at least 50% or higher of ARR reduction for a DMT to be categorized as high-efficacy therapy. You provide references 22 and 23 on this subject, but not the original which explains clearly the reason for this classification.  Perhaps a brief commentary would also be in order explaining why DMF is not classify as a HE DMT despite its trial DEFINE showed 53% reduction of Relapse Rate.  

Author Response

We thank the reviewers for their time and effort. We carefully considered their comments and tried to address all raised issues. We think the review is greatly improved thanks to the suggested revisions.

You provide a comprehensive and extensive (this latter observation is discussed with the Editor), review of this practical and actuality topic of global interest. I have a couple of minor commentaries on your manuscript. 

1) In my opinion, a comment should be made on the Association of British Neurologists Guidelines expressed in 2015, regarding at least 50% or higher of ARR reduction for a DMT to be categorized as high-efficacy therapy. You provide references 22 and 23 on this subject, but not the original which explains clearly the reason for this classification.  Perhaps a brief commentary would also be in order explaining why DMF is not classify as a HE DMT despite its trial DEFINE showed 53% reduction of Relapse Rate. 

Response: thank you for permitting us to clarify this aspect. The 2015 ABN guidelines suggest that agents can be divided into two broad classes: drugs of moderate efficacy (average relapse reduction in 30–50% range), including the β-interferons, glatiramer, teriflunomide, dimethyl fumarate and fingolimod; and drugs of high efficacy (average relapse reduction substantially more than 50%), alemtuzumab and natalizumab.

In these guidelines, DMF was not considered a HE-DMT. Indeed, although DMF demonstrated an ARR reduction of 53% in the DEFINE trial, it also showed an ARR reduction of 44% in the CONFIRM study. These percentages do, therefore, not satisfy the adopted “substantially more than 50%” criterion of a HE-DMT. It is true, though, that fingolimod was also considered a moderate-efficacy drug, while we included it in the HE group in the present review.

Since the ABN guidelines were published in 2015, substantial new data on efficacy has been gathered, both for S1P-modulators other than fingolimod and for newer agents (such as cladribine anti-CD20 agents). This evidence (summarized in the cited network meta-analysis by Samjoo et al, 2021 and in the Expert Opinion by Filippi et al, 2022 – refs 22 and 23) categorize drugs as HE-DMT based not only on ARR (but also on other outcomes such as MRI activity and the effect on disability progression), and based on updated evidence. Accordingly, fingolimod, together with alemtuzumab, natalizumab, ocrelizumab, ofatumumab, ozanimod and cladribine (the latter 4 were not included in the ABN 2015 guidelines) are now considered HE-DMT, with fingolimd and ozanimod showing a comparable a ARR reduction compared to cladribine.  

We have, nevertheless, acknowledged the 2015 ABN guidelines, and the following sentence has been modified as follows: "Considering their efficacy, particularly concerning relapse reduction rate [23], DMTs are generally distinguished into moderate efficacy (ME) DMTs, including Interferon-beta (IFN-β), Dimethyl fumarate, Glatiramer acetate, Teriflunomide, and high efficacy (HE) DMTs, encompassing Ocrelizumab, Ofatumumab, Alemtuzumab, Cladribine, Natalizumab and Sphingosine-1 phosphate modulators [22], although previously published guidelines included the Sphingosine-1 phosphate modulator fingolimod among ME DMTS".

Round 2

Reviewer 1 Report

Comments and Suggestions for Authors

The authors improved the manuscript significantly. Regarding the aspect of hypogammaglobulinemia I would still suggest to include the very important and comprehensive analysis of Derfuss et all. Ther Adv Neurol Disord. 2024 Oct 8

Author Response

The authors improved the manuscript significantly. Regarding the aspect of hypogammaglobulinemia I would still suggest to include the very important and comprehensive analysis of Derfuss et all. Ther Adv Neurol Disord. 2024 Oct 8.

Response:

Thank you very much for the appreciation of our efforts.

We have included the suggested recent reference and added the following brief sentence on the study conclusions in the manuscript:

A more recent pooled post-hoc long-term analysis of interventional trials and their open-label extension studies concluded that time on ocrelizumab and abnormal IgG levels were not significantly associated with an increased SI risk, but, rather, abnormal IgM levels. However, authors acknowledge the possibility of attrition bias and a limited generalizability to real-world settings [103]. (Section 2.1.2, subsection Infections)